# Learning Ordinal Probabilistic Reward from Preferences

**Longze Chen**[1,2]  **Lu Wang**[3]  **Renke Shan**[3]  **Ze Gong**[1]*  **Run Luo**[1,2]  **Jiaming Li**[1,2]
**Jing Luo**[1]  **Qiyao Wang**[1,2]  **Min Yang**[1,4]*
[1] Shenzhen Institutes of Advanced Technology, Chinese Academy of Sciences
[2] University of Chinese Academy of Sciences    [3] Ritzz-AI
[4] Shenzhen University of Advanced Technology
{lz.chen2, ze.gong, min.yang}@siat.ac.cn
 https://github.com/ritzz-ai/OPRM

## ABSTRACT

Reward models are crucial for aligning large language models (LLMs) with human values and intentions. Existing approaches follow either Generative (GRMs) or Discriminative (DRMs) paradigms, yet both suffer from limitations: GRMs typically demand costly point-wise supervision, while DRMs produce uncalibrated relative scores that lack probabilistic interpretation. To address these challenges, we introduce a novel reward modeling paradigm: ***Probabilistic Reward Model*** (PRM). Instead of modeling reward as a deterministic scalar, our approach treats it as a random variable, learning a full probability distribution for the quality of each response. To make this paradigm practical, we present its closed-form, discrete realization: the ***Ordinal Probabilistic Reward Model*** (OPRM), which discretizes the quality score into a finite set of ordinal ratings. Building on OPRM, we propose a data-efficient training strategy called ***Region Flooding Tuning*** (RgFT). It enables rewards to better reflect absolute text quality by incorporating quality-level annotations, which guide the model to concentrate the probability mass within corresponding rating sub-regions. Experiments on various reward model benchmarks show that our method improves accuracy by **2.9%~7.4%** compared to prior reward models, demonstrating strong performance and data efficiency. Analysis of the score distribution provides evidence that our method captures not only relative rankings but also absolute quality.

## 1 INTRODUCTION

Reinforcement Learning from Human Feedback (RLHF) has emerged as a pivotal technique for aligning Large Language Models (LLMs) with human values and intentions (Achiam et al., 2023; Ouyang et al., 2022). As a critical component of the RLHF process (Bai et al., 2022), the reward model is trained to assign scores that quantify the degree of alignment between the model's outputs and human preferences. Recent advances (Guo et al., 2025a; Lightman et al., 2024) have shown that well-designed reward signals, whether applied during training or inference, can significantly enhance LLM performance across diverse domains (Shao et al., 2024; Luo et al., 2025; Jin et al., 2025; Wang et al., 2025b; Li et al., 2025). However, learning a reward model that can accurately capture human preference signals remains a significant challenge (Gao et al., 2023; Sun et al., 2025a; Zhong et al., 2025). Most recent efforts typically follow either the generative or discriminative paradigm, yet both approaches exhibit inherent limitations that hinder their effectiveness in practice.

*Discriminative Reward Models* (DRMs), which append an MLP-based value head to a base model, are commonly optimized with the Bradley-Terry objective to output a scalar reward (Liu et al., 2024b; Cai et al., 2024; Lou et al., 2024). A key limitation of this paradigm is that its reward scores reflect only relative preferences, not intrinsic quality. It indicates that one response is preferred but fails to explain why, making it difficult to establish a trusted acceptance threshold to discern high-quality responses from low-quality ones. In response, *Generative Reward Models* (GRMs) have emerged (Mahan

---

*Corresponding authors.

Table 1: **Comparison of OPRM with baseline reward models across multiple dimensions**. Margin Sensitivity (whether distinguish samples with subtle preference differences), Require Training (whether requires training on preference data), Value Head Free (whether eliminates the need for additional value head), and Input Flexibility (whether supports rating single and multiple responses).

| Baselines | Input Format | Output Format | Margin Sensitivity | Require Training | Value Head Free | Input Flexible |
|---|---|---|---|---|---|---|
| Bradley-Terry (Bradley & Terry, 1952) | Single Response | Continuous Score | ✓ | ✓ | ✗ | ✓ |
| PairRM (Jiang et al., 2023) | Response Pairs | Continuous Score | ✓ | ✓ | ✗ | ✗ |
| CLoud (Ankner et al., 2024) | Single Response | Critique + Continuous Score | ✓ | ✓ | ✗ | ✓ |
| LLM-as-a-Judge (Zheng et al., 2023) | Response Pairs | Discrete Score | ✗ | ✗ | ✓ | ✗ |
| Pointwise GRM (Liu et al., 2025b) | Single Response | Critique + Discrete Score | ✗ | ✓ | ✓ | ✓ |
| **OPRM (Ours)** | Single Response | Continuous Score | ✓ | ✓ | ✓ | ✓ |

et al., 2024; Zhang et al., 2024). These models leverage the native generative capabilities of LLMs to produce Chain-of-Thought critiques before rendering a preference judgment, conceptually aligning with the LLM-as-a-Judge paradigm (Zheng et al., 2023). While GRMs offer superior interpretability through their critique generation, they often rely on rigid pairwise input formats that limit flexibility in Best-of-N (BoN) scenarios. Moreover, achieving performance comparable to DRMs frequently requires costly pointwise supervision for calibration, substantially increasing the annotation burden. Consequently, the field faces a critical trade-off: choosing between the efficiency of DRMs and the interpretability of GRMs, with neither approach offering a complete solution.

To transcend this trade-off, we introduce a novel reward modeling paradigm: ***Probabilistic Reward Model*** (PRM). Instead of approximating rewards with a deterministic scalar value like the Bradley-Terry model (Bradley & Terry, 1952), PRM reframes the task as learning a full probability distribution over the reward space. Since learning this continuous distribution is computationally intractable, we translate it into a discrete realization: ***Ordinal Probabilistic Reward Model*** (OPRM). Specifically, OPRM discretizes the reward space into a finite set of ordinal ratings (Liu et al., 2025a; Wang et al., 2025a), thereby replacing the intractable integration with a closed-form summation that makes our paradigm more practical. Thus, OPRM resolves the core trade-off. By providing a full reward distribution, it unlocks richer interpretability and uncertainty estimation than DRMs, while its flexible input format enables efficient scoring of single or multiple responses, making it better suited than GRMs for modern evaluations like Best-of-N (BoN). Table 1 summarizes the advantages of OPRM over existing reward modeling baselines.

Building upon the OPRM paradigm, we further propose ***Region Flooding Tuning*** (RgFT), a novel training strategy designed to calibrate the reward distribution to reflect absolute textual quality. The core principle of RgFT is to leverage quality-level annotations (i.e., **good**, **normal**, and **bad**) on preference data. Rather than optimizing over the full distribution of ordinal ratings, RgFT guides the model to concentrate probability mass within rating sub-regions corresponding to the quality-level labels. While simply restricting scores to specific quality intervals can lead to optimization stagnation due to constant gradients, RgFT **floods** these rigid constraints into a triangular probability landscape. This restores the gradient incentives, guiding the model to not only locate the correct quality region but also maximize the preference margin by pushing scores towards the extremes of their respective ranges. Critically, RgFT facilitates semi-supervised training by jointly leveraging a mixture of quality-labeled and preference-only data, obviating the need for costly large-scale annotation.

Before delving into details, we summarize our contributions as follows:

- We propose a novel reward modeling paradigm, the Ordinal Probabilistic Reward Model. By learning a full probability distribution for a response's quality, OPRM mitigates the core trade-off between the efficiency of DRMs and the interpretability of GRMs.

- We design a data-efficient training strategy, Region Flooding Tuning, which grounds the reward distribution in an absolute quality scale by guiding the model to concentrate probability mass within correct rating sub-regions.

- We conduct extensive experiments on four benchmarks covering over ten tasks, demonstrating the effectiveness of OPRM in precise reward modeling across diverse scenarios. Additional studies confirm that RgFT significantly improves the accuracy, robustness, and interpretability of OPRM.

## 2 RELATED WORK

**Discriminative Reward Model.** Discriminative reward model typically consists of a base model and a MLP-based reward head (classifier) that outputs a scalar score for a given input. These models are commonly trained using the Bradley-Terry (BT) (Bradley & Terry, 1952) loss to maximize the reward margin between chosen and rejected responses. While the core BT loss remains a standard component, considerable research has focused on enhancing data quality and refining the modeling framework (e.g. Skywork-reward (Liu et al., 2024b), InternLM2-reward (Cai et al., 2024), Helpsteer2-preference (Wang et al., 2024b), QRM (Dorka, 2024), URM (Lou et al., 2024), CLoud (Ankner et al., 2024), ArmoRM (Wang et al., 2024a), and PURM (Sun et al., 2025b)), further boosting DRM performance. Nonetheless, these methods are limited to learning a pairwise ranking, yielding scores that are unbounded and difficult to interpret. In contrast, our approach learns a probabilistic distribution over scores, which enables more reliable and calibrated outputs.

**Generative Reward Model.** Generative reward models directly leverage LLM-generated outputs to evaluate preference, which is aligned with the LLM-as-a-Judge paradigm (Zheng et al., 2023). These models output chain-of-thought reasoning (critiques) before generating preference judgments (e.g., Critic-RM (Yu et al., 2024), PROMETHEUS (Kim et al., 2023), CLoud (Ankner et al., 2024), GenRM (Zhang et al., 2024; Mahan et al., 2024), Synthetic Critique (Ye et al., 2024), and RISE (Yu et al., 2025)), enhancing the interpretability of the reward signals. Recent advances have employed reinforcement learning to construct reasoning-based reward models (SPCT (Liu et al., 2025b), RM-R1 (Chen et al., 2025b), J1 (Whitehouse et al., 2025), RRM (Guo et al., 2025b), and JudgeLRM (Chen et al., 2025a)), demonstrating promising scalability in inference-time computation. However, these approaches often struggle to outperform DRMs under computational constraints. Conversely, our approach achieves comparable efficiency and performance while preserving interpretability.

**Ordinal Regression and Distribution Learning.** Deep ordinal regression has evolved from simple continuous discretization (Fu et al., 2018; Rothe et al., 2018) to distribution ordering learning (Wang et al., 2025a). Prominent methods like SORD (Diaz & Marathe, 2019), ALDL (Li et al., 2022), and POE (Li et al., 2021) model label distributions or latent uncertainty to capture ordinal relationships effectively. While OPRM draws inspiration from these probabilistic frameworks to construct reward distributions over the LLM vocabulary, it is conceptually distinct from RLHF approaches utilizing ordinal feedback (Liu et al., 2025a). Prior RLHF works typically focus on refining the granularity of input supervision (e.g., "significantly better") for continuous regressors. In contrast, OPRM enforces ordinality within the output representation space, treating rewards as discrete variables on a statistically ordinal scale anchored to semantic quality, thereby bridging distributional ordinal regression with pairwise preference optimization.

## 3 PRELIMINARIES

**Preference data annotation.** To annotate the preference data, the SFT model $\pi^{\text{SFT}}$ is given prompts $x$ to two distinct outputs $(y_1, y_2) \sim \pi^{\text{SFT}}(y \mid x)$. These output pairs are then presented to human labelers, who express their preference for one output. This preference can be denoted as $y_c \succ y_r \mid x$, where $y_c$ and $y_r$ represent the chosen and rejected outputs, respectively, from the pair $(y_1, y_2)$.

**Standard Bradley-Terry Reward Modeling.** Following the Bradley-Terry model (Bradley & Terry, 1952), we model the probability of preferring response $y_c$ over $y_r$ based on their underlying scalar rewards, which are provided by a reward function $r_\psi(x, y)$. This preference distribution is formulated as follows:

$$P_\psi(y_c \succ y_r \mid x) = \frac{\exp\left(r_\psi(x, y_c)\right)}{\exp\left(r_\psi(x, y_c)\right) + \exp\left(r_\psi(x, y_r)\right)},$$
$$= \sigma(r_\psi(x, y_c) - r_\psi(x, y_r)), \tag{1}$$

which $\sigma$ is the logistic function. Treating the problem as a binary classification task yields the negative log-likelihood loss function:

$$\mathcal{L}(r_\psi) = -\mathbb{E}_{(x, y_c, y_r) \sim \mathcal{D}_{\text{rm}}}[\log P_\psi(y_c \succ y_r \mid x)],$$
$$= -\mathbb{E}_{(x, y_c, y_r) \sim \mathcal{D}_{\text{rm}}}[\log \sigma(r_\psi(x, y_c) - r_\psi(x, y_r))], \tag{2}$$

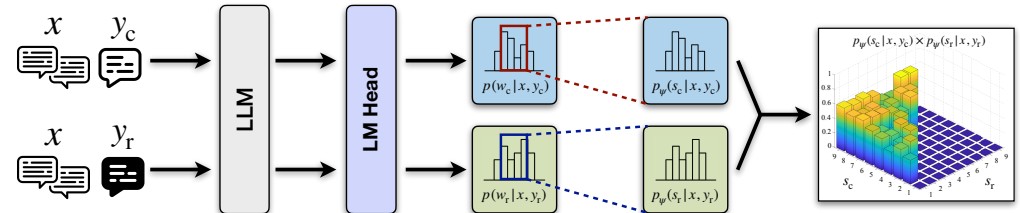

Figure 1: **The architectures of Ordinal Probabilistic Reward Model**. Given a problem and a pair of responses, designated as **chosen** and **rejected**, the OPRM utilizes its language model (LM) head to obtain the ordinal rating probabilities for each response. A joint probability matrix is then constructed by computing the Cartesian product of these two sets of probabilities for optimization.

where dataset is composed of comparisons denoted as $\mathcal{D}_{\text{rm}} = \{x^{(i)}, y_{\text{c}}^{(i)}, y_{\text{r}}^{(i)}\}_{i=1}^{N}$. In the realm of LLMs, the network $r_\psi(x, y)$ is often initialized using the SFT model $\pi^{\text{SFT}}(y \mid x)$. It then incorporates an additional linear layer on the final transformer layer to generate a singular scalar prediction representing the reward value.

## 4 ORDINAL PROBABILISTIC REWARD MODEL

In this section, we introduce **Ordinal Probabilistic Reward Model**, a novel reward modeling paradigm that learns a probability distribution over response quality. We begin by outlining the continuous form of our reward modeling optimization paradigm, the **Probabilistic Reward Modeling** (§ 4.1). We then discretize this formulation into a tractable form, termed OPRM (§ 4.2) and conclude by presenting the complete training and inference pipeline for OPRM (§ 4.3).

### 4.1 PROBABILISTIC REWARD MODELING

Departing from the conventional Bradley-Terry reward model (Section 3), which estimates a single scalar value for each response, we propose a reward modeling objective derived from Random Utility Model theory (Manski, 1977; Cascetta, 2009). Our objective enables the model to learn a probability distribution over the quality of each response. Concretely, we model the quality score of a response $y$ for a given input $x$ as a continuous random variable $S$. This random variable is supported on a bounded interval $[a, b] \subset \mathbb{R}$. Our reward model, parameterized by $\psi$, learns the conditional probability density function (PDF) $p_\psi(s \mid x, y)$ of this variable, where $s$ is a realization of $S$. This density must satisfy $\int_a^b p_\psi(s \mid x, y) \, ds = 1$.

Given a preference pair $(y_{\text{c}}, y_{\text{r}})$ with a chosen and a rejected response, we model their quality scores as two independent random variables, $S_{\text{c}}$ and $S_{\text{r}}$. Their scores are drawn from the distributions defined by their respective conditional PDFs: $s_{\text{c}} \sim p_\psi(\cdot \mid x, y_{\text{c}})$ and $s_{\text{r}} \sim p_\psi(\cdot \mid x, y_{\text{r}})$. The probability of the preference $y_{\text{c}} \succ y_{\text{r}}$ is then modeled as the probability that the score of the chosen response exceeds that of the rejected one:

$$P_\psi(y_{\text{c}} \succ y_{\text{r}} \mid x) = \mathbb{E}_{s_{\text{c}}, s_{\text{r}}} [\mathbb{1}(s_{\text{c}} > s_{\text{r}})] \tag{3}$$

where $\mathbb{1}(\cdot)$ denotes the indicator function. Expanding the expectation in integral form over the bounded interval $[a, b]$, we obtain:

$$P_\psi(y_{\text{c}} \succ y_{\text{r}} \mid x) = \int_a^b \int_a^b \mathbb{1}(s_{\text{c}} > s_{\text{r}}) \, p_\psi(s_{\text{c}} \mid x, y_{\text{c}}) \, p_\psi(s_{\text{r}} \mid x, y_{\text{r}}) \, ds_{\text{r}} \, ds_{\text{c}} \tag{4}$$

This expression corresponds to computing the probability that a random score sampled from the chosen response exceeds a random score from the rejected response, integrating over their joint distribution constrained to the bounded interval. Since $\mathbb{1}(s_{\text{c}} > s_{\text{r}}) = 1$ only when $s_{\text{c}} > s_{\text{r}}$ and $0$ otherwise, so it effectively truncates the integral domain to $(a, s_{\text{c}})$ for $p_\psi(s_{\text{r}} \mid x, y_{\text{r}})$. Thus, we can equivalently restructure the double integral as follows:

$$P_\psi(y_{\text{c}} \succ y_{\text{r}} \mid x) = \int_a^b p_\psi(s_{\text{c}} \mid x, y_{\text{c}}) \left( \int_a^{s_{\text{c}}} p_\psi(s_{\text{r}} \mid x, y_{\text{r}}) \, ds_{\text{r}} \right) ds_{\text{c}} \tag{5}$$

Figure 2: **Region Flooding Tuning**. To ensure the correctness of the reward modeling, region flooding is applied to each of the three partition combinations, resulting in a lower triangular form.

Finally, we can simply optimize the Eq. (5) by minimizing the negative log-likelihood loss. Notably, the Bradley-Terry model is a special case of the Probabilistic Reward Modeling framework, arising when the quality score distribution is constrained to a unimodal Gumbel distribution with fixed shape parameters (see Appendix B for a detailed proof). However, this objective lacks a closed-form analytical solution and requires estimation through Monte Carlo sampling. This computational challenge motivates our transition from the continuous formulation to a more tractable discrete one.

## 4.2 FROM CONTINUOUS TO DISCRETE

To obtain a closed-form analytical solution, we adapt the continuous formulation in Eq. (5) by modeling the scores as discrete random variables over a finite set of ordinal ratings $\{a, a + 1, \ldots, b\}$. This yields the following closed-form expression:

$$P_\psi(y_{\mathrm{c}} \succ y_{\mathrm{r}} \mid x) = \sum_{s_{\mathrm{c}}=a}^{b} p_\psi(s_{\mathrm{c}} \mid x, y_{\mathrm{c}}) \left( \sum_{s_{\mathrm{r}}=a}^{s_{\mathrm{c}}-1} p_\psi(s_{\mathrm{r}} \mid x, y_{\mathrm{r}}) \right) \tag{6}$$

As observed in Eq. (2), the Bradley-Terry model maximizes the score gap between chosen $(y_{\mathrm{c}})$ and rejected $(y_{\mathrm{r}})$ responses, creating a steep reward landscape with pronounced gradients beneficial for RL optimization. Similarly, our optimization objective in Eq. (6) inherits and generalizes this desirable property. By operating over full reward distributions instead of single scalars, our objective naturally shifts probability mass upward for chosen responses and downward for rejected ones, thereby widening their separation.

This intuition is formally supported by the gradient dynamics of the probability mass functions (PMFs). Specifically, the sensitivity of the objective $J \triangleq P_\psi(y_{\mathrm{c}} \succ y_{\mathrm{r}} \mid x)$ with respect to the mass at score $k$ is given by Eq. (7).

$$\frac{\partial J}{\partial p_{\mathrm{c}}(k)} = P(s_{\mathrm{r}} < k), \quad \frac{\partial J}{\partial p_{\mathrm{r}}(k)} = P(s_{\mathrm{c}} > k). \tag{7}$$

These derivatives imply that increasing the probability mass for $y_{\mathrm{c}}$ at score $k$ is incentivized whenever $y_{\mathrm{r}}$ is likely to be lower than $k$. Consequently, shifting mass from a lower score $k$ to a higher score $k + 1$ for the chosen response yields a strictly non-negative gain proportional to $p_{\mathrm{r}}(k)$. This creates a consistent optimization pressure driving the distribution of $y_{\mathrm{c}}$ towards the maximum score $b$ and $y_{\mathrm{r}}$ towards the minimum score $a$. A detailed proof via gradient analysis can be found in Appendix C.

**Ordinal Probabilistic Reward Modeling** presents two key advantages: (1) *Quantifying Uncertainty*, the variance of the output distribution serves as a measure of model confidence—wide distributions for ambiguous comparisons indicate uncertainty, while sharp, peaked distributions reflect clear preferences, enhancing interpretability. Our method thus explicitly captures the inherent uncertainty in human preference judgments, a crucial aspect often overlooked by discriminative reward models. (2) *Handling Annotation Disagreement*, our method can represent multimodal score distributions (e.g., Mixture of Gaussians), enabling it to capture disagreements among annotators. By explicitly capturing conflicting signals within the score distribution, our model becomes robust to the performance degradation often caused by inconsistent preference data (Sun et al., 2025a). This contrasts sharply with traditional methods like the Bradley-Terry model, which are restricted to unimodal preferences.

## 4.3 PIPELINE

Our training pipeline, illustrated in Figure 1, begins by formatting the preference data pairs $(x, y_{\mathrm{c}}, y_{\mathrm{r}})$ (see Section 3) into a structured input using a prompt template. The details of this template and criteria

Table 2: Overall results of different methods and models on four RM benchmarks. **bold numbers** indicate the best performance. Underlined numbers indicate the second best. The Overall* score is the average performance excluding Reward Bench due to its known data contamination issues.

| Model | Reward Bench | PPE-P | PPE-C | RMB | Overall | Overall* |
|---|---|---|---|---|---|---|
| *Reported Results of Public Models* | | | | | | |
| Skywork-Reward-Gemma-2-27B | 93.8 | 56.6 | 56.6 | 60.2 | 66.8 | 57.8 |
| DeepSeek-V2.5-0905 | 81.5 | 62.8 | 58.5 | 65.7 | 67.1 | 62.3 |
| Gemini-1.5-Pro | 86.8 | 66.1 | 59.8 | 56.5 | 67.3 | 60.8 |
| ArmoRM-8B-v0.1 | 90.4 | 60.6 | 61.2 | 64.6 | 69.2 | 62.1 |
| InternLM2-20B-Reward | 90.2 | 61.0 | 63.0 | 62.9 | 69.3 | 62.3 |
| LLaMA-3.1-70b-Instruct | 84.1 | 65.3 | 59.2 | 68.9 | 69.4 | 64.5 |
| Claude-3.5-sonnet | 84.2 | 65.3 | 58.8 | 70.6 | 69.7 | 63.2 |
| Nemotron-4-340B-Reward | 92.0 | 59.3 | 60.8 | 69.9 | 70.5 | 63.3 |
| GPT-4o | 86.7 | 67.1 | 57.6 | 73.8 | 71.3 | 66.2 |
| *Reproduced Results of Baseline Methods From DeepSeek* | | | | | | |
| LLM-as-a-Judge | 83.4 | 64.2 | 58.8 | 64.8 | 67.8 | 62.6 |
| DeepSeek-BTRM-27B | 81.7 | 68.3 | 66.7 | 57.9 | 68.6 | 64.3 |
| CLoud-Gemma-2-27B | 82.0 | 67.1 | 62.4 | 63.4 | 68.7 | 64.3 |
| DeepSeek-PairRM-27B | 87.1 | 65.8 | 64.8 | 58.2 | 69.0 | 62.9 |
| DeepSeek-GRM-27B-RFT | 84.5 | 64.1 | 59.6 | 67.0 | 68.8 | 63.6 |
| DeepSeek-GRM-27B | 86.9 | 64.7 | 59.8 | 69.0 | 69.9 | 64.5 |
| *Results of Our Method* | | | | | | |
| **OPRM-Qwen2.5-7B** | 87.8 | 61.1 | 61.3 | 71.5 | 70.4 | 64.6 |
| **OPRM-Qwen2.5-14B** | 89.3 | 63.0 | 64.3 | 73.8 | 72.6 | 67.0 |
| **OPRM-Qwen2.5-32B** | 91.3 | 63.9 | 66.1 | 75.6 | **74.2** | 68.5 |
| **OPRM-Qwen2.5-72B** | 89.3 | 65.1 | 64.3 | 73.5 | 73.1 | 67.6 |
| *Results of Our Method (w/ Region Flooding Tuning)* | | | | | | |
| **OPRM-RgFT-Qwen2.5-7B** | 86.2 | 62.3 | 62.4 | 70.1 | 70.3(↓0.1) | 64.9(↑0.3) |
| **OPRM-RgFT-Qwen2.5-14B** | 87.3 | 63.4 | 65.6 | 72.8 | 72.3(↓0.3) | 67.3(↑0.3) |
| **OPRM-RgFT-Qwen2.5-32B** | 88.9 | 64.6 | 67.3 | 74.8 | 73.9(↓0.3) | **68.9**(↑0.4) |
| **OPRM-RgFT-Qwen2.5-72B** | 89.1 | 65.3 | 66.4 | 74.2 | 73.8(↑0.7) | 68.6(↑1.0) |

are provided in Appendix H. As the next step in the pipeline, following the parameter-free technique from prior work (Cui et al., 2023), we compute the distribution over quality score $s \in \{a, a+1, \ldots, b\}$ (where $a, b \in \mathbb{Z}$) by directly repurposing the LM head's vocabulary probabilities, thus obviating the need for a separate prediction head and avoiding any new parameters. In our implementation, we set the quality score range from 1 to 9 (i.e., $a = 1, b = 9$) The score distribution is then formed by directly extracting the vocabulary probabilities of the corresponding numeric tokens (i.e., '1' to '9'). This approach allows the model to directly leverage its inherent ordinal knowledge of numbers.

In summary, both the chosen and rejected inputs are fed into the LLM backbone and its LM head, yielding the post-softmax vocabulary probability distributions $p(w_c \mid x, y_c)$ and $p(w_r \mid x, y_r)$ at the last token position. The probabilities of all numeric tokens are then normalized to form the distribution for our ordinal probabilistic reward modeling:

$$p_\psi(s_c = i \mid x, y_c) = \frac{p(w_c = \text{`i'} \mid x, y_c)}{\sum_{j=1}^9 p(w_c = \text{`j'} \mid x, y_c)} , \ p_\psi(s_r = i \mid x, y_r) = \frac{p(w_r = \text{`i'} \mid x, y_r)}{\sum_{j=1}^9 p(w_r = \text{`j'} \mid x, y_r)}$$
(8)

Finally, we substitute the obtained $p_\psi(s_c = i \mid x, y_c)$ and $p_\psi(s_r = i \mid x, y_r)$ into Eq. (6) and maximize $P_\psi(y_c \succ y_r \mid x)$ using the negative log-likelihood loss. See Appendix L for a detailed computational overhead analysis.

During the inference stage, we simply input a response $y$ given prompt $x$ to obtain a quality score distribution $p_\psi(s \mid x, y)$. We can derive a scalar reward score through either argmax or weighted averaging. For a discussion of other possible decoding strategies, see Appendix K.2. In our subsequent experiments, we adopt the straightforward weighted averaging approach to compute the reward score: $r_\psi(x, y) = \sum_{s=a}^b s \cdot p_\psi(s \mid x, y)$, avoiding the tie-prone argmax method.

Table 3: Detailed results of different methods on the PPE Correctness benchmark. **Bold numbers** indicate the best performance. Underlined numbers indicate the second best.

| Model | MMLU-Pro | MATH | GPQA | MBPP-Plus | IFEval | PPE Correctness |
|---|---|---|---|---|---|---|
| *Results of Our Method* | | | | | | |
| OPRM-Qwen2.5-7B | 65.2 | 70.1 | 56.3 | 59.0 | 56.1 | 61.3 |
| OPRM-Qwen2.5-14B | 66.7 | 70.7 | 57.1 | 67.4 | 59.5 | 64.3 |
| OPRM-Qwen2.5-32B | 71.2 | 73.2 | 57.9 | 66.2 | 62.2 | 66.1 |
| OPRM-Qwen2.5-72B | 73.4 | 75.9 | 58.6 | 54.1 | 59.5 | 64.3 |
| *Results of Our Method (w/ Region Flooding Tuning)* | | | | | | |
| **OPRM-RgFT-Qwen2.5-7B** | 64.8 | 71.6 | 55.9 | 63.0 | 56.8 | 62.4$_{(\uparrow 1.1)}$ |
| **OPRM-RgFT-Qwen2.5-14B** | 69.5 | 74.0 | 57.3 | 67.0 | 60.0 | 65.6$_{(\uparrow 1.3)}$ |
| **OPRM-RgFT-Qwen2.5-32B** | 73.3 | 76.8 | 58.5 | 67.2 | 60.6 | **67.3**$_{(\uparrow 1.2)}$ |
| **OPRM-RgFT-Qwen2.5-72B** | 72.8 | 77.1 | 59.0 | 62.0 | 61.2 | 66.4$_{(\uparrow 2.1)}$ |

## 5 REGION FLOODING TUNING

While OPRM effectively captures relative preferences, precisely aligning its scoring distribution with absolute quality judgments presents a further challenge. To address this, we introduce **Region Tuning** (RgT), a cost-effective method that enhances the model's fidelity to absolute quality scores using minimal annotations (§ 5.1). Subsequently, we refine RgT to preserve the desirable properties (as detailed in Appendix C), culminating in our final method: **Region Flooding Tuning** (RgFT) (§ 5.2).

### 5.1 REGION TUNING

Building upon the OPRM optimization objective from Eq. (6), which employs the finite set of ordinal ratings $S = \{1, 2, \ldots, 9\}$ for all data, we introduce a more fine-grained partitioning based on the absolute quality of each response, a technique we term ***Region Tuning*** (RgT).

Specifically, we further partition the finite set into three quality levels, guiding the model to concentrate the probability mass within corresponding rating sub-region: $S_{bad} = \{1, 2, 3\}$, $S_{normal} = \{4, 5, 6\}$, and $S_{good} = \{7, 8, 9\}$. Consequently, for a single preference data point consisting of a chosen and a rejected response, there are six possible combinations of quality levels. These include pairs from different levels, as well as pairs where both responses fall into the same level, denoted as <$l_{\text{chosen}}, l_{\text{rejected}}$>: <**good**, **normal**>, <**good**, **bad**>, <**normal**, **bad**>, <**good**, **good**>, <**normal**, **normal**>, <**bad**, **bad**>.

This partitioning allows us to redefine the preference probability by conditioning it on the quality levels of the chosen and rejected responses. Thus, the optimization objective is formulated as:

$$P_\psi(y_c \succ y_r \mid x, l_{\text{chosen}}, l_{\text{rejected}}) = \sum_{s_c \in S_{l_{\text{chosen}}}} p_\psi(s_c \mid x, y_c) \left( \sum_{s_r \in S_{l_{\text{rejected}}}} p_\psi(s_r \mid x, y_r) \mathbb{1}(s_c > s_r) \right) \tag{9}$$

Details on the partition of the semantic regions ($S_{\text{bad}}$, $S_{\text{normal}}$, $S_{\text{good}}$) are provided in Appendix E.

### 5.2 FROM REGION TUNING TO REGION FLOODING TUNING

As shown in Figure 2, when $l_{\text{chosen}} \neq l_{\text{rejected}}$, Eq. (9) optimizes a square-shaped joint probability region, resulting in constant partial derivatives $\frac{\partial P}{\partial p_c(k)}$ and $\frac{\partial P}{\partial p_r(k)}$. In this case, the optimization objective no longer shifts the probability mass of the chosen response upwards and the rejected response downwards to increase their separation. This leads to the loss of a desirable property of OPRM, as mentioned in Section 4.2 (see Appendix C and G for a formal proof and analysis).

As shown in Figure 2, we propose region flooding to the optimized joint probability region, expanding it into a lower triangular shape to preserve the desired property. As its expansion process closely resembles breadth-first search algorithm, we term it ***Region Flooding Tuning*** (RgFT). RgFT provides three key advantages: (1) *Interpretability*, RgFT constrains the model to concentrate probability mass within the score regions correspond to pre-defined quality levels, enabling reward scores to more accurately reflect the absolute quality of responses. (2) *Semi-supervised Learning*, RgFT supports semi-supervised training by combining quality-labeled data with preference-only data. (3)

Table 4: Detailed results of Qwen2.5-7B with different methods on the Role Play benchmark. **Bold numbers** indicate the best performance. Underlined numbers indicate the second best.

| Method | Pair-Accuracy | Best-of-N | Best-of-N-plus | Worst-of-N | Overall |
|---|---|---|---|---|---|
| Random Baseline | 50.0 | 25.5 | 31.3 | 68.7 | 43.9 |
| *Training on Role Play Data Only* | | | | | |
| BT Model | 70.4 | 48.6 | 51.3 | 83.6 | 63.5 |
| BT Model - w/ Margin | 71.0 | 49.3 | 52.3 | 84.2 | 64.2 |
| **OPRM** (ours) | 71.3 | 49.4 | 52.5 | 84.1 | 64.3 |
| **OPRM-RgFT** (ours) | 72.1($\uparrow$0.8) | 50.7($\uparrow$1.3) | 53.6($\uparrow$1.1) | 85.1($\uparrow$1.0) | 65.4($\uparrow$1.1) |
| *Training on Mixed Role Play and General-Domain Data* | | | | | |
| BT Model | 73.8 | 51.2 | 54.3 | 86.0 | 66.3 |
| BT Model - w/ Margin | 75.3 | 53.4 | 55.7 | 87.2 | 67.9 |
| **OPRM** (ours) | 74.4 | 54.1 | 56.1 | 87.8 | 68.1 |
| **OPRM-RgFT** (ours) | **75.8**($\uparrow$1.4) | **55.8**($\uparrow$1.7) | **59.3**($\uparrow$3.2) | **89.9**($\uparrow$2.1) | **70.2**($\uparrow$2.1) |

*Customizability*, RgFT allows for the flexible tailoring of rating sub-regions to their corresponding quality levels, making the strategy adaptable to diverse application requirements (see Appendix K.1).

# 6 EXPERIMENTS

## 6.1 EXPERIMENTAL SETUP

In our experiments, we curate a dataset of 130k samples for reward model training, drawn primarily from publicly available open-source datasets: **Skywork Reward Preference 80K** (Liu et al., 2024b) and **UltraFeedback Binarized Preferences** (Cui et al., 2023). We employ the Qwen2.5-Instruction series of models (7B, 14B, 32B, and 72B) (Team, 2024) as the backbone for training the OPRM. We compare OPRM to different categories of baselines: **Discriminative RMs**, **Generative RMs** and **DeepSeek-RM**. Following prior work, we evaluate the performance of different methods on various RM benchmarks: **Reward Bench** (Lambert et al., 2024), **PPE-Preference**, **PPE-Correctness** (Frick et al., 2024), and **RMB** (Zhou et al., 2024). We use the standard pair accuracy and Best-of-N evaluation metrics for each benchmark. Detailed information on the training preference data, baselines, benchmarks, and evaluation metrics is provided in Appendix D.

## 6.2 MAIN RESULTS

As shown in Table 2, we compares the overall results of OPRM with different baseline reward models on RM benchmarks. We present the performance of OPRM with the reported results of public models and the reproduced results of baseline methods from DeepSeek. We observe that OPRM outperforms the baseline methods in overall performance, and achieves competitive results against strong public RMs, such as Nemotron-4-340B-Reward and GPT-4o. Notably, the 14B, 32B, and 72B models surpass all prior leading reward models, improving upon the previous best result by **1.3%**, **2.9%**, and **1.8%**, respectively, despite being significantly smaller in scale. Moreover, the most significant performance enhancement is observed on the RMB and PPE-Correctness benchmarks, which utilize Best-of-N evaluation to better reflect practical effectiveness on downstream tasks. We attribute the 32B model's superior performance over the 72B model to the exceptional zero-shot capability of Qwen2.5-32B. This enables it to outperform larger models on the RM benchmark without any fine-tuning. The more detailed numbers on RewardBench, PPE Correctness, and RMB are in Table 8, Table 9, and Table 10 in Appendix I.

## 6.3 THE IMPACT OF REGION FLOODING TUNING

Building upon OPRM, we incorporate RgFT for further experimentation. We train the OPRM-RgFT series of models using the preference data, further enriched with three defined quality level annotations (**good**, **normal**, and **bad**). Consistent with the advantages of RgFT described in Section 5.

### 6.3.1 FEWER ANNOTATIONS, BETTER RESULTS

As presented in Table 2 and Table 3, our evaluation of OPRM-RgFT on four RM benchmarks reveals a notable performance divergence. On one hand, RgFT consistently improves performance across all model sizes on the PPE benchmarks. Notably, OPRM-RgFT-32B achieves SOTA accuracy of 67.3% on the PPE-Correctness benchmark, surpassing all prior leading reward models. On the other hand, its performance on other benchmarks is inconsistent. We hypothesize that this discrepancy stems from biases introduced by our annotation strategy for general data (see Appendix J). This process, involving coarse AI annotation with simple manual correction, is effective for verifiable tasks with explicit correctness labels like PPE-Correctness but likely introduces label noise for other tasks. Further supporting this claim, our subsequent experiments show that incorporating fine-grained manual annotations leads to consistent performance improvements.

### 6.3.2 BEYOND ACCURACY, RELIABLE CALIBRATION

While high accuracy is desirable, a reliable reward model must also provide calibrated confidence, particularly for probabilistic approaches. To strictly quantify this, we measure the Expected Calibration Error (ECE) on the Reward-Bench dataset. Given the inherent noise in fine-grained ordinal ratings, we aggregate scores into semantic tiers (**bad**, **normal**, **good**) to compute

Table 5: Comparison of accuracy and calibration (ECE-10) on the RewardBench.

| Model | Acc (%) | ECE-10 (%, ↓) |
|---|---|---|
| Qwen-2.5-32B | 69.56 | 26.72 |
| OPRM-32B | 81.04 | 10.62 |
| **OPRM-RgFT-32B** | **90.90** | **5.18** |

a robust ECE against ground truth labels verified by GPT-4o and humans. As shown in Table 5, OPRM-32B significantly outperforms the baseline, reducing ECE by over 60%. Crucially, OPRM-RgFT-32B achieves a minimal ECE of 5.18%, an 80.6% relative reduction compared to the baseline. This demonstrates that RgFT not only boosts ranking performance but also effectively regularizes probability estimates, preventing overconfidence and ensuring alignment with empirical correctness.

### 6.3.3 TOWARDS HUMAN-ALIGNED SCORE DISTRIBUTIONS.

To evaluate the impact of Region Flooding Tuning on absolute quality assessment, we analyze the score distributions produced by our models. Specifically, we curated two distinct datasets for this analysis: an **Absolute-Good Set** with 100 high-quality prompt-response pairs and an **Absolute-Bad Set** with 100 poor-quality pairs. These pairs are manually selected by experts based on a multi-faceted evaluation across dimensions such as instruction following, factual accuracy, and helpfulness. We then score both datasets using three models: the baseline BTRM-32B, our base model OPRM-32B, and its RgFT-enhanced version. As illustrated in Figure 3, the base OPRM-32B already exhibits a basic capacity for absolute quality assessment: within its $[1, 9]$ scoring range, it generally assigns scores above 5 to good responses and below 5 to bad ones. Crucially, OPRM-RgFT-32B significantly enhances this capability. The RgFT-enhanced model polarizes the score

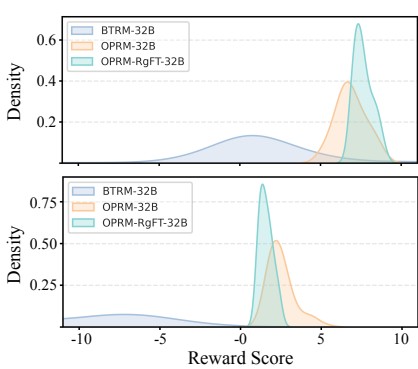

Figure 3: Comparison of score distributions for responses of high-quality (**Top**) and low-quality (**Bottom**).

distributions, pushing scores for high-quality responses into the $[7, 9]$ range while confining low-quality ones to $[1, 3]$. This increased separation makes the score itself a more reliable and interpretable indicator of absolute quality. Case studies in Appendix M provide detailed scoring examples that further corroborate these findings and demonstrate the improved reliability of RgFT scores.

### 6.3.4 SEMI-SUPERVISED DOMAIN ADAPTATION

To simulate practical applications and reduce annotation costs, we investigate RgFT's effectiveness in a semi-supervised domain adaptation setting. Specifically, we curated a training set of 31K role-playing instances with quality-level labels (see Appendix J for annotation details) and a mixed dataset by combining these with an equal volume of unlabeled general-domain preference data.

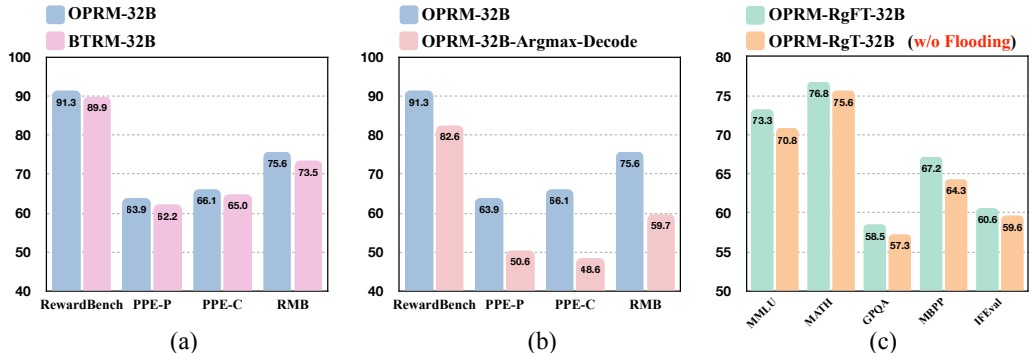

Figure 4: **Ablation Study**: (a) Assessing the superiority of OPRM over the BT Model. (b) Evaluating the efficacy of Weighted Average Decoding. (c) Validating the necessity of Region Flooding.

For evaluation, we build a test set of 500 questions, each with 5-10 responses, and designed three new metrics: **Best-of-N** (top-scoring is **good**-level), **Worst-of-N** (bottom-scoring is **bad**-level), and **Best-of-N-plus**(top-scoring is not **bad**-level). As shown in Table 4, we benchmark our models against BT and BT-with-Margin baselines (see Appendix F for detailed formulas) under two settings: training on role-play data only, and on the mixed dataset. In both settings, OPRM surpasses the baselines, and OPRM-RgFT further improves upon OPRM. Crucially, incorporating unlabeled general-domain data significantly boosts the performance of OPRM and OPRM-RgFT from $64.3\%$ to $68.1\%$ and $65.4\%$ to $70.2\%$, respectively. This demonstrates that RgFT can effectively leverage unlabeled preference data in a semi-supervised manner, offering a cost-effective path for domain adaptation.

## 6.4 ABLATION STUDY

We perform an ablation study to isolate and evaluate the key components of our OPRM and RgFT frameworks. As depicted in Figure 4, the removal of any single component detrimentally affects the overall performance.

**Effectiveness of OPRM Loss.** Replacing the proposed OPRM loss with the standard Bradley-Terry (BT) loss on an identical Qwen-32B model incurs a consistent performance penalty of $1.1\%$ to $2.1\%$ across all benchmarks (Figure 4(a)). This result empirically substantiates the advantage of modeling the reward as an ordinal variable rather than a strict pairwise preference.

**Impact of Decoding Method.** We contrast our weighted averaging decoding strategy with a naive Argmax baseline, which greedily selects the highest-probability token. As shown in Figure 4(b), Argmax leads to a substantial $8.7\%$ to $17.5\%$ performance drop. This is primarily because Argmax fails to capture fine-grained quality differences, thereby generating an excessive number of ties.

**Necessity of the Flooding Mechanism.** The flooding mechanism is designed to create desirable lower triangular score regions (see Appendix C). Removing it resulted in a $1.0\%$ to $2.9\%$ performance drop on the PPE Correctness benchmark (see Figure 4(c)). The degradation is most pronounced when distinguishing between marginally different responses, confirming the mechanism's critical role.

## 7 CONCLUSION

In this paper, we propose ***Ordinal Probabilistic Reward Model***, a novel paradigm that learns a full probability distribution over an ordinal reward space. To better anchor these rewards to absolute quality, we further proposed ***Region Flooding Tuning***, a training strategy that leverages quality-level annotations to calibrate the model's probability distribution. Extensive experiments on four diverse reward modeling benchmarks show that our approach consistently improves performance by $2.9\%$ to $7.4\%$. Furthermore, detailed analysis reveals that OPRM is superior to the conventional Bradley-Terry model and that RgFT is crucial for discerning fine-grained quality differences. We believe OPRM with RgFT offer a powerful framework for developing more accurate and reliable reward models, a critical step towards building more capable and aligned large language models.

ACKNOWLEDGMENTS

Min Yang was supported by National Key Research and Development Program of China (2024YFF0908200), National Natural Science Foundation of China (Grant No. 62376262), and Natural Science Foundation of Guangdong Province of China (2024A1515030166, 2025B1515020032).

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

APPENDIX

## A  THE USE OF LARGE LANGUAGE MODELS (LLMS)

We utilize Large Language Models (LLMs) to aid in the writing and polishing of this manuscript. Specifically, LLMs are employed to correct grammatical errors, improve sentence structure, and enhance the clarity and conciseness of the text. This process is primarily applied to the Introduction, Related Work, and Appendix sections. All scientific contributions, methodologies, and conclusions presented in this paper are the original work of the authors. The LLMs serve solely as a writing-enhancement tool.

## B  BRADLEY-TERRY AS A SPECIAL CASE OF PROBABILISTIC REWARD MODELING

In this section, we demonstrate that the Bradley-Terry model for pairwise preferences can be derived from a more general probabilistic reward modeling framework under a specific set of distributional assumptions.

Let $p_\psi(s \mid x, y)$ denote the probability density function of a score $s$ assigned to a response $y$ given a context $x$, where the scoring mechanism is parameterized by $\psi$. Consider two responses for the same context $x$: a chosen response $y_c$ and a rejected response $y_r$. Let $s_c$ and $s_r$ be the random variables for their respective scores, with distributions $p_\psi(s_c \mid x, y_c)$ and $p_\psi(s_r \mid x, y_r)$. We assume $s_c$ and $s_r$ are conditionally independent given $x, y_c, y_r$.

The probability that $y_c$ is preferred over $y_r$, denoted $P_\psi(y_c \succ y_r \mid x)$, is the probability that the score of the chosen response is greater than that of the rejected one, i.e., $P(s_c > s_r)$. This can be expressed generally as:

$$P_\psi(y_c \succ y_r \mid x) = \int_{-\infty}^{\infty} p_\psi(s_c \mid x, y_c) \left( \int_{-\infty}^{s_c} p_\psi(s_r \mid x, y_r) \, \mathrm{d}s_r \right) \mathrm{d}s_c. \tag{10}$$

We show that this general formulation reduces to the Bradley-Terry model under specific assumptions. For clarity, we first establish the functional form of the Bradley-Terry model in terms of the sigmoid function.

**Lemma B.1** (Bradley-Terry Model in Sigmoid Form). *The Bradley-Terry (BT) model, which defines the preference probability based on underlying quality scores $r_\psi(x, y_c)$ and $r_\psi(x, y_r)$ as*

$$P_{BT}(y_c \succ y_r \mid x) = \frac{\exp(r_\psi(x, y_c))}{\exp(r_\psi(x, y_c)) + \exp(r_\psi(x, y_r))}, \tag{11}$$

*is equivalent to the sigmoid function of the difference in scores:*

$$P_{BT}(y_c \succ y_r \mid x) = \sigma(r_\psi(x, y_c) - r_\psi(x, y_r)), \tag{12}$$

*where $\sigma(z) = 1/(1 + e^{-z})$ is the standard logistic sigmoid function.*

*Proof.* We start from the standard definition of the BT model and manipulate it algebraically. By dividing the numerator and the denominator of Eq. (equation 11) by $\exp(r_\psi(x, y_r))$, we obtain:

$$\begin{aligned}
P_{BT}(y_c \succ y_r \mid x) &= \frac{\exp(r_\psi(x, y_c)) \cdot \exp(-r_\psi(x, y_r))}{(\exp(r_\psi(x, y_c)) + \exp(r_\psi(x, y_r))) \cdot \exp(-r_\psi(x, y_r))} \\
&= \frac{\exp(r_\psi(x, y_c) - r_\psi(x, y_r))}{\exp(r_\psi(x, y_c) - r_\psi(x, y_r)) + 1} \\
&= \frac{\exp(r_\psi(x, y_c) - r_\psi(x, y_r))}{1 + \exp(r_\psi(x, y_c) - r_\psi(x, y_r))}.
\end{aligned}$$

To bring this into the form of the sigmoid function $\sigma(z)$, we can divide the numerator and denominator by $\exp(r_\psi(x, y_\mathrm{c}) - r_\psi(x, y_\mathrm{r}))$:

$$
\begin{aligned}
P_{\mathrm{BT}}(y_\mathrm{c} \succ y_\mathrm{r} \mid x) &= \frac{1}{\frac{1 + \exp(r_\psi(x,y_\mathrm{c}) - r_\psi(x,y_\mathrm{r}))}{\exp(r_\psi(x,y_\mathrm{c}) - r_\psi(x,y_\mathrm{r}))}} \\
&= \frac{1}{\exp(-(r_\psi(x, y_\mathrm{c}) - r_\psi(x, y_\mathrm{r}))) + 1} \\
&= \sigma(r_\psi(x, y_\mathrm{c}) - r_\psi(x, y_\mathrm{r})).
\end{aligned}
$$

This completes the proof of the lemma. $\qquad\square$

With this lemma, we can prove the main proposition.

**Proposition B.2.** *The general preference probability $P_\psi(y_\mathrm{c} \succ y_\mathrm{r} \mid x)$ defined in Eq. (10) is equivalent to the Bradley-Terry model if the following assumptions hold:*

1. *The score difference $\Delta s_\psi \triangleq s_\mathrm{c} - s_\mathrm{r}$ follows a logistic distribution.*

2. *The mean of this logistic distribution is the difference of deterministic underlying quality scores: $\mu = r_\psi(x, y_\mathrm{c}) - r_\psi(x, y_\mathrm{r})$.*

3. *The scale parameter of the logistic distribution is unity ($s = 1$).*

*Proof.* The preference probability is the probability that the score of the chosen response exceeds that of the rejected one. This can be expressed in terms of the score difference random variable $\Delta s_\psi = s_\mathrm{c} - s_\mathrm{r}$:

$$
P_\psi(y_\mathrm{c} \succ y_\mathrm{r} \mid x) = P(s_\mathrm{c} > s_\mathrm{r}) = P(\Delta s_\psi > 0). \tag{13}
$$

**Assumption 1** states that $\Delta s_\psi$ follows a logistic distribution. The cumulative distribution function (CDF) of a logistic random variable $Z$ with mean $\mu$ and scale $s$ is given by $F_Z(z) = (1 + e^{-(z-\mu)/s})^{-1}$. Therefore, we can compute the preference probability as:

$$
\begin{aligned}
P(\Delta s_\psi > 0) &= 1 - P(\Delta s_\psi \le 0) \\
&= 1 - F_{\Delta s_\psi}(0) \\
&= 1 - \frac{1}{1 + e^{-(0-\mu)/s}} \\
&= 1 - \frac{1}{1 + e^{\mu/s}} \\
&= \frac{(1 + e^{\mu/s}) - 1}{1 + e^{\mu/s}} = \frac{e^{\mu/s}}{1 + e^{\mu/s}} \\
&= \frac{1}{1 + e^{-\mu/s}}.
\end{aligned} \tag{14}
$$

This final expression is precisely the sigmoid function, $\sigma(\mu/s)$.

Then, we apply the remaining assumptions. **Assumption 2** posits that the mean of the distribution is $\mu = r_\psi(x, y_\mathrm{c}) - r_\psi(x, y_\mathrm{r})$. **Assumption 3** sets the scale parameter to unity, $s = 1$. Substituting these into our result from Eq. (14) yields:

$$
P_\psi(y_\mathrm{c} \succ y_\mathrm{r} \mid x) = \frac{1}{1 + e^{-(r_\psi(x,y_\mathrm{c}) - r_\psi(x,y_\mathrm{r}))}} = \sigma(r_\psi(x, y_\mathrm{c}) - r_\psi(x, y_\mathrm{r})). \tag{15}
$$

From Lemma B.1, we know that the Bradley-Terry model also simplifies to $\sigma(r_\psi(x, y_\mathrm{c}) - r_\psi(x, y_\mathrm{r}))$. Since the probabilistic reward model under the specified assumptions and the Bradley-Terry model both yield the identical functional form, we have shown that the latter is a special case of the former. $\qquad\square$

# C GRADIENT ANALYSIS OF THE PREFERENCE PROBABILITY

In this section, we conduct a formal gradient-based analysis to demonstrate that maximizing the preference probability, $P_\psi(y_c \succ y_r \mid x)$, incentivizes the underlying probabilistic model to maximally separate the score distributions of the chosen and rejected responses.

**Proposition C.1** (Optimization Incentive of Preference Maximization). *Let the scores for responses be drawn from a discrete set $\{a, a+1, \ldots, b\}$. Let $p_c(k) \triangleq p_\psi(s_c = k \mid x, y_c)$ and $p_r(k) \triangleq p_\psi(s_r = k \mid x, y_r)$ be the respective probability mass functions (PMFs). Maximizing the preference probability $P(s_c > s_r)$ with respect to the variables $\{p_c(k)\}$ and $\{p_r(k)\}$ under the constraints $\sum_k p_c(k) = 1$ and $\sum_k p_r(k) = 1$ creates the following incentives:*

1. *For the chosen response $y_c$, shifting probability mass from any score $k$ to a higher score $k+1$ will increase or maintain the objective value.*

2. *For the rejected response $y_r$, shifting probability mass from any score $k+1$ to a lower score $k$ will increase or maintain the objective value.*

*This implies that the optimization process drives the PMF of $y_c$ towards the maximum score $b$ and the PMF of $y_r$ towards the minimum score $a$.*

*Proof.* The preference probability $P \triangleq P_\psi(y_c \succ y_r \mid x)$ for discrete scores is given by:

$$P = \sum_{i=a}^{b} p_c(i) P(s_r < i) = \sum_{i=a}^{b} p_c(i) \left( \sum_{j=a}^{i-1} p_r(j) \right). \tag{16}$$

We analyze the gradient of $P$ with respect to the probability mass at each score for $y_c$ and $y_r$ separately.

**Part 1: Incentive for the Chosen Response Score ($y_c$).** We first compute the partial derivative of $P$ with respect to $p_c(k)$ for some score $k \in \{a, \ldots, b\}$. From Eq. (16), only the term where $i = k$ depends on $p_c(k)$, so:

$$\frac{\partial P}{\partial p_c(k)} = \frac{\partial}{\partial p_c(k)} \left[ p_c(k) \sum_{j=a}^{k-1} p_r(j) \right] = \sum_{j=a}^{k-1} p_r(j) = P(s_r < k). \tag{17}$$

This derivative represents the sensitivity of the objective to an increase in probability mass at score $k$. To understand the incentive for shifting mass, consider moving an infinitesimal probability mass $\epsilon > 0$ from a score $k$ to a higher score $k+1$. This corresponds to a change in the PMF: $p_c(k) \to p_c(k) - \epsilon$ and $p_c(k+1) \to p_c(k+1) + \epsilon$. The resulting change in $P$, denoted $\Delta P$, can be approximated by the first-order Taylor expansion (which is exact since $P$ is linear in $p_c$):

$$\begin{aligned}
\Delta P &\approx \epsilon \frac{\partial P}{\partial p_c(k+1)} - \epsilon \frac{\partial P}{\partial p_c(k)} \\
&= \epsilon \left( P(s_r < k+1) - P(s_r < k) \right) \quad \text{(using Eq. equation 17)} \\
&= \epsilon \cdot P(s_r = k) \\
&= \epsilon \cdot p_r(k).
\end{aligned} \tag{18}$$

Since probabilities are non-negative, $p_r(k) \geq 0$, and we defined $\epsilon > 0$, it follows that $\Delta P \geq 0$. This demonstrates that any shift of probability mass to a higher score for $y_c$ is guaranteed to be a non-decreasing change in the objective function. This creates a persistent optimization pressure to move the entire distribution $p_c$ towards the maximum score $b$.

**Part 2: Incentive for the Rejected Response Score ($y_r$).** To analyze the effect of $p_r(k)$, it is more convenient to rewrite Eq. (16) by swapping the order of summation:

$$P = \sum_{j=a}^{b-1} p_r(j) \left( \sum_{i=j+1}^{b} p_c(i) \right). \tag{19}$$

The partial derivative of $P$ with respect to $p_{\mathrm{r}}(k)$ for $k \in \{a, \ldots, b-1\}$ is:

$$\frac{\partial P}{\partial p_{\mathrm{r}}(k)} = \frac{\partial}{\partial p_{\mathrm{r}}(k)} \left[ p_{\mathrm{r}}(k) \sum_{i=k+1}^{b} p_{\mathrm{c}}(i) \right] = \sum_{i=k+1}^{b} p_{\mathrm{c}}(i) = P(s_{\mathrm{c}} > k). \qquad (20)$$

Now, consider shifting an infinitesimal probability mass $\epsilon > 0$ from a score $k+1$ to a *lower* score $k$. This corresponds to the change: $p_{\mathrm{r}}(k) \to p_{\mathrm{r}}(k) + \epsilon$ and $p_{\mathrm{r}}(k+1) \to p_{\mathrm{r}}(k+1) - \epsilon$. The resulting change in $P$ is:

$$\begin{aligned}
\Delta P &\approx \epsilon \frac{\partial P}{\partial p_{\mathrm{r}}(k)} - \epsilon \frac{\partial P}{\partial p_{\mathrm{r}}(k+1)} \\
&= \epsilon \left( P(s_{\mathrm{c}} > k) - P(s_{\mathrm{c}} > k+1) \right) \quad \text{(using Eq. equation 20)} \\
&= \epsilon \cdot P(s_{\mathrm{c}} = k+1) \\
&= \epsilon \cdot p_{\mathrm{c}}(k+1).
\end{aligned} \qquad (21)$$

Since $p_{\mathrm{c}}(k+1) \geq 0$ and $\epsilon > 0$, we have $\Delta P \geq 0$. This shows that shifting probability mass to a lower score for $y_{\mathrm{r}}$ is always a non-decreasing change. This creates a consistent optimization pressure to move the distribution $p_{\mathrm{r}}$ towards the minimum score $a$.

Combining both parts, we have formally shown that maximizing the preference probability $P(s_{\mathrm{c}} > s_{\mathrm{r}})$ drives the model to separate the score distributions by pushing the mass of $p_{\mathrm{c}}$ towards the highest possible score and the mass of $p_{\mathrm{r}}$ towards the lowest possible score. $\qquad \square$

## D    DETAILED EXPERIMENTAL SETUP

**Training Preference Data.**    We curate a dataset of 130k samples for reward model training, drawn primarily from publicly available open-source datasets: **Skywork Reward Preference 80K** (Liu et al., 2024b) is a high-quality, pairwise preference dataset that spans multiple domains, including chat, safety, mathematics, and code. It employs advanced data filtering techniques to ensure the reliability of preferences across different tasks. **UltraFeedback Binarized Preferences** (Cui et al., 2023) is a large-scale, fine-grained, and diverse preference dataset designed for training powerful reward and critic models. It comprises approximately 64k prompts from various sources, including UltraChat (Ding et al., 2023), ShareGPT, Evol-Instruct (Xu et al., 2024), TruthfulQA (Lin et al., 2021), FalseQA (Hu et al., 2023), and FLAN (Wei et al., 2021). Each prompt is used to query multiple LLMs to generate four distinct responses, resulting in a total of 256k samples.

**Baselines.**    In our main experiments, we employ the Qwen2.5-Instruction series of models (7B, 14B, 32B, and 72B) (Team, 2024) as the backbone for training the OPRM. We compare OPRM to different categories of baselines: (1) **Discriminative RMs**, including Skywork-Reward (Liu et al., 2024b), ArmoRM (Wang et al., 2024a), InternLM-20B-Reward (Cai et al., 2024), and Nemotron-4-340B-Reward (Wang et al., 2024c). (2) **Generative RMs**, including DeepSeek-V2.5 (Liu et al., 2024a), Gemini-1.5-Pro (Team et al., 2024), LLaMA-3.1-70B (Grattafiori et al., 2024), Claude-3.5-sonnet, and GPT-4o (Hurst et al., 2024). (3) **DeepSeek-RM**, a collection of baselines re-implemented by DeepSeek, including LLM-as-A-Judge (Zheng et al., 2023), DeepSeek-BTRM (Bradley & Terry, 1952), DeepSeek-PairRM (Jiang et al., 2023), CLoud-Gemma-2 (Ankner et al., 2024) and DeepSeek-GRM (Liu et al., 2025b).

**Benchmarks and Evaluation Metrics.**    Following prior work, we evaluate the performance of different methods on various RM benchmarks of different domains: **Reward Bench** (Lambert et al., 2024), **PPE-Preference**, **PPE-Correctness** (Frick et al., 2024), **RMB** (Zhou et al., 2024). We use the standard Best-of-N evaluation metrics for each benchmark: the accuracy of picking the best response from a set of responses. Specifically, Reward Bench and PPE Preference involve pairwise comparisons, with each prompt featuring two candidate responses. In contrast, PPE Correctness is designed for a large-scale Best-of-N evaluation, presenting 32 responses for each prompt. RMB is a hybrid, incorporating both pairwise comparison tasks and a Best-of-5 selection format.

# E    SENSITIVITY ANALYSIS ON BOUNDARY AND BIN CONFIGURATIONS

In this section, we investigate the robustness of OPRM-RgFT to variations in ordinal bin definitions and boundary placements. Specifically, we aim to verify that the method's performance is primarily driven by the underlying probabilistic framework rather than specific hyperparameter choices regarding the ordinal scale.

To this end, we trained a variant of OPRM-RgFT-32B using an expanded scale $S' = \{0, \ldots, 9\}$ with *irregular boundaries*: **bad** $\{0, 1, 2, 3\}$, **normal** $\{4, 5\}$, and **good** $\{6, 7, 8, 9\}$. This setup introduces asymmetry and changes the cardinality of the semantic sets, contrasting with our default uniform configuration which employs a symmetric partition of the scale $S = \{1, \ldots, 9\}$ (**bad** $\{1, 2, 3\}$, **normal** $\{4, 5, 6\}$, **good** $\{7, 8, 9\}$).

As presented in Table 6, the performance deviation between the default uniform setting and the irregular variant is negligible, with the difference in the Overall score being less than $0.1\%$. This empirical evidence demonstrates that OPRM-RgFT is robust to boundary shifts and does not rely on uniform partitions to achieve high performance.

Table 6: Performance comparison between the default uniform configuration and an irregular boundary variant. The results indicate that the method is highly robust to binning strategies.

| Configuration | RewardBench | PPE-P | PPE-C | RMB | Overall |
|---|---|---|---|---|---|
| **Irregular Variant** (Scale $0 \ldots 9$) | | | | | |
| bad $\{0, 1, 2, 3\}$, normal $\{4, 5\}$, good $\{6, 7, 8, 9\}$ | 89.1 | 64.1 | 67.7 | 74.4 | 73.8 |
| **Default Uniform** (Scale $1 \ldots 9$) | | | | | |
| bad $\{1, 2, 3\}$, normal $\{4, 5, 6\}$, good $\{7, 8, 9\}$ | 88.9 | 64.6 | 67.3 | 74.8 | 73.9 |

Despite the demonstrated robustness to irregular boundaries, we adhere to the default uniform configuration for two principled reasons related to efficiency and priors. First, regarding **computational efficiency**, we utilize the range $[1, 9]$ to maximize granularity while ensuring each ordinal score maps to a single token. Mainstream LLM tokenizers typically treat digits $\{0, \ldots, 9\}$ as individual tokens, whereas values $\geq 10$ are decomposed into multiple tokens. Restricting the support set to single tokens allows OPRM to compute the full probability distribution in a single forward pass, avoiding the prohibitive computational costs associated with computing joint probabilities over multi-token sequences. Second, concerning **geometric simplicity**, we employ a uniform partition ($3 \times 3$ regions) as a neutral prior to minimize inductive bias. This uniformity aligns with the theoretical design of Region Flooding Tuning, allowing probability mass to flood into a symmetric lower triangular geometry of consistent size. In the absence of domain-specific knowledge suggesting that **good** samples require finer granularity than **bad** ones, a symmetric division simplifies hyperparameter selection and ensures balanced gradient pressure across different quality levels.

# F    BRADLEY-TERRY LOSS WITH MARGIN

Inspired by INF-ORM (Yang et al., 2024), which employs GPT-4o to evaluate the preference margin between chosen and rejected responses, we annotate each pair in our dataset with a margin label. The original evaluation in INF-ORM follows these rules: (1) If the chosen answer is much better than rejected answer, set margin to 10; (2) If the chosen answer is better than the rejected answer, set margin to 3; (3) If the chosen answer is slightly better than rejected answer, set margin to 1.

Analogously, we define margins based on the combination of quality-level annotations. Specifically, pairs with the same quality level, such as <**good**, **good**>, <**normal**, **normal**>, and <**bad**, **bad**>, are assigned a margin of 1. Pairs with adjacent quality levels, namely <**good**, **normal**> and <**normal**, **bad**>, are assigned a margin of 3. Finally, a margin of 10 is assigned to pairs with distant quality levels, like <**good**, **bad**>.

After that, the Bradley-Terry Loss with Margin is defined as:

$$\mathcal{L}(r_\psi) = -\mathbb{E}_{(x, y_c, y_r) \sim \mathcal{D}_{rm}}[m(x, y_c, y_r) \cdot \log \sigma(r_\psi(x, y_c) - r_\psi(x, y_r))], \tag{22}$$

Here, $m(x, y_c, y_r)$ stands for the margin value between chosen and rejected responses. This formula helps the model to better understand which responses are preferred over others, based on the scores we gave them.

## G  DISENTANGLING THE IMPACT OF REGION FLOODING FROM LABEL AUGMENTATION

A central hypothesis of this work is that the *Region Flooding Tuning* (RgFT) framework provides methodological benefits beyond the simple inclusion of absolute quality labels. To verify whether the observed performance gains stem from the architecture itself or merely from data augmentation, we conducted controlled experiments comparing OPRM-RgFT against standard classification paradigms trained on identical data (comprising both preference pairs and semantic quality labels).

We formulate two baseline approaches to isolate the contribution of the modeling strategy:

- **Baseline A (Hard Classification):** A standard multi-class classification model optimizing for three discrete quality tiers (**good**, **normal**, **bad**). Inference is performed via maximum a posteriori estimation, selecting the class with the highest probability.

- **Baseline B (Scalar-Weighted Classification):** A regression-oriented approach designed to provide finer granularity than hard classification. Here, the reward score is computed as the expected value over class probabilities: $\mathbb{E}[s] = \sum_{c \in \{\text{good}, \text{normal}, \text{bad}\}} P(c) \cdot v_c$, where $v_c$ represents the centroid of the corresponding semantic region (e.g., values mapped to 2, 5, and 8).

Table 7: Comparative analysis of OPRM-RgFT against standard classification baselines utilizing identical supervision signals. The results demonstrate that the proposed region flooding mechanism yields significant gains over pure classification approaches.

| Method | RewardBench | PPE-P | PPE-C | RMB | Overall |
|---|---|---|---|---|---|
| Baseline A (Hard Classification) | 71.0 | 43.9 | 46.6 | 55.7 | 54.3 |
| Baseline B (Scalar-Weighted Classification) | 85.4 | 61.5 | 64.3 | 71.3 | 70.6 |
| **OPRM-RgFT-32B (Ours)** | **88.9** | **64.6** | **67.3** | **74.8** | **73.9** |

The empirical results, detailed in Table 7, indicate a clear performance hierarchy. Hard Classification significantly outperforms Scalar-Weighted Classification, underscoring the necessity of continuous score representations for ranking tasks. However, it consistently underperforms OPRM-RgFT across all benchmarks (e.g., a 3.3% deficit in the Overall score). This performance gap substantiates that the efficacy of our method is not solely attributable to the availability of absolute labels. Rather, the Region Flooding strategy effectively harmonizes discriminative pairwise ranking with absolute semantic constraints. Unlike standard classification, which treats labels as independent categories, RgFT anchors the preference distribution within a continuous ordinal space, thereby achieving superior calibration and ranking fidelity.

## H  OPRM'S PROMPT TEMPLATES

As shown in Prompt H.1, we present the prompt templates used during the OPRM training process. Furthermore, we designed distinct scoring criteria for general-domain and domain-specific data. Specifically, the general-domain preference data is evaluated using the OpenAI 3H Criteria, as detailed in Prompt H.2. In contrast, the role-play data used in our experiments is assessed against our custom Role Play Criteria, presented in Prompt H.2.

## H.1   PROMPT

> **Ordinal Distributional Reward Modeling Prompt**
>
> ## Evaluation Guidelines
> You are an expert text quality evaluator.
> Your task is to carefully assess and score the AI-generated candidate response based on the specific criteria provided below.
>
> ## Scoring Criteria
>
> {criteria}
>
> ## Context of Interaction
> The following represents a historical conversation between a user and an AI assistant:
>
> {prompt}
>
> ## AI Response to Evaluate
> Below is the candidate response generated by the AI that needs to be evaluated:
>
> {response}
>
> ## Evaluation Instructions
> Please score the response according to the criteria on a scale from 1 to 9, where:
> - 1-3: Bad performance
> - 4-6: Normal performance
> - 7-9: Good performance
>
> Use the following format for your final assessment: "Score: <total points>".
>
> Score:

## H.2   CRITERIA

> **Role Play Criteria**
>
> **Relevant**: Responses should not be irrelevant. Responses should be coherent and reasonable within the given context, without containing logical flaws.
>
> **Credible**:   Responses should be credible, meet the prompt requirements and provide accurate information.  In role-playing or storytelling, responses should be consistent with the given character and background.
>
> **Interesting**:   Responses should be engaging and able to capture the user's attention. Responses that are self-aware, thought-provoking, or captivating will be more interesting.

> **Openai 3H Criteria**
>
> **Helpfulness**: The content generated should aim to be of assistance to the user. It should provide accurate information, answer questions to the best of the model's knowledge, and offer solutions that are practical and relevant to the user's queries. The model should not generate content that is intentionally misleading or unhelpful.
>
> **Truthfulness**:   The model should strive to provide information that is truthful and based on the most accurate data available up to its knowledge cutoff. It should not fabricate information or present opinions as facts.  When uncertain, the model should acknowledge the limitations of its knowledge.
>
> **Harmlessness**: The content should not cause harm to individuals or groups. It should avoid generating content that is offensive, discriminatory, or promotes harmful behaviors.  The model should be programmed to refuse to generate content that could be used for malicious purposes, including but not limited to generating false information, engaging in deception, or promoting illegal activities.

# I DETAILED EXPERIMENT RESULTS

We report the detailed per-subset experiment results on RewardBench 8, PPE Correctness 9, and RMB 10. The results for the baseline methods are sourced from their original papers.

Table 8: Detailed results of different methods on the Reward Bench benchmark.

| Method | Chat | Chat Hard | Safety | Reasoning | Reward Bench |
|---|---|---|---|---|---|
| *Reported Results of Public Models* | | | | | |
| Skywork-Reward-Gemma-2-27B | 95.8 | 91.4 | 91.9 | 96.1 | 93.8 |
| DeepSeek-V2.5-0905 | - | - | - | - | 81.5 |
| Gemini-1.5-Pro | 94.1 | 77.0 | 85.8 | 90.2 | 86.8 |
| ArmoRM-8B-v0.1 | 96.9 | 76.8 | 90.5 | 97.3 | 90.4 |
| InternLM2-20B-Reward | 98.9 | 76.5 | 89.5 | 95.8 | 90.2 |
| LLaMA-3.1-70b-Instruct | 97.2 | 70.2 | 82.8 | 86.0 | 84.1 |
| Claude-3.5-sonnet | 96.4 | 74.0 | 81.6 | 84.7 | 84.2 |
| Nemotron-4-340B-Reward | 95.8 | 87.1 | 91.5 | 93.6 | 92.0 |
| GPT-4o | 96.1 | 76.1 | 88.1 | 86.6 | 86.7 |
| *Reproduced Results of Baseline Methods From DeepSeek* | | | | | |
| LLM-as-a-Judge | 96.7 | 69.3 | 83.5 | 84.3 | 83.4 |
| DeepSeek-BTRM-27B | 96.7 | 86.2 | 75.7 | 89.8 | 81.7 |
| CLoud-Gemma-2-27B | 96.7 | 69.3 | 83.5 | 84.3 | 82.0 |
| DeepSeek-PairRM-27B | 95.5 | 86.8 | 52.3 | 92.0 | 87.1 |
| DeepSeek-GRM-27B-RFT | 94.7 | 77.2 | 87.0 | 79.2 | 84.5 |
| DeepSeek-GRM-27B | 94.1 | 78.3 | 88.0 | 83.8 | 86.0 |
| *Results of Our Method* | | | | | |
| **OPRM-Qwen2.5-7B** | 96.4 | 76.3 | 86.2 | 92.2 | 87.8 |
| **OPRM-Qwen2.5-14B** | 96.6 | 78.1 | 86.1 | 96.2 | 89.3 |
| **OPRM-Qwen2.5-32B** | 96.9 | 81.8 | 89.6 | 96.7 | 91.3 |
| **OPRM-Qwen2.5-72B** | 96.4 | 79.6 | 88.1 | 93.0 | 89.3 |
| *Results of Our Method (w/ Region Flooding Tuning)* | | | | | |
| **OPRM-RgFT-Qwen2.5-7B** | 95.5 | 76.5 | 86.4 | 86.5 | 86.2 |
| **OPRM-RgFT-Qwen2.5-14B** | 96.9 | 79.4 | 88.1 | 84.6 | 87.3 |
| **OPRM-RgFT-Qwen2.5-32B** | 95.3 | 82.7 | 89.2 | 88.4 | 88.9 |
| **OPRM-RgFT-Qwen2.5-72B** | 96.9 | 82.7 | 89.7 | 87.1 | 89.1 |

# J QUALITY-LEVEL ANNOTATION

## J.1 GENERAL-DOMAIN DATA

For the acquisition of quality-level annotations, we follow the methodology of UltraFeedback (Cui et al., 2023). The process involves two main steps. First, we employ the gpt-4o model (Hurst et al., 2024) to annotate each prompt-response pair with fine-grained scores across multiple dimensions, such as instruction-following, truthfulness, and helpfulness. Second, these scores are averaged, and the resulting value is mapped to one of our three predefined quality levels—**good**, **normal**, or **bad**—based on specific score intervals.

Following this automatic annotation, we perform a manual verification step. For verifiable tasks, such as mathematics and coding, we check the responses against the ground truth. If a response is found to be incorrect, its quality level is manually downgraded to **bad**. We acknowledge that for more subjective tasks, this per-instance verification is not always feasible, which may introduce some annotation noise.

Finally, to ensure logical consistency, we filter out all pairs where the chosen response is not strictly better than the rejected one. This includes invalid combinations of $<l_{\text{chosen}}, l_{\text{rejected}}>$ such as <**normal**, **good**>, <**bad**, **normal**>, and <**bad**, **good**>. The remaining data constitutes our final training set.

Table 9: Detailed results of different methods on the PPE Correctness benchmark.

| Method | MMLU-Pro | MATH | GPQA | MBPP-Plus | IFEval | PPE Correctness |
|---|---|---|---|---|---|---|
| *Reported Results of Public Models* | | | | | | |
| Skywork-Reward-Gemma-2-27B | 54.0 | 63.0 | 53.0 | 59.0 | 54.0 | 56.6 |
| DeepSeek-V2.5-0905 | - | - | - | - | - | 58.5 |
| Gemini-1.5-Pro | - | - | - | - | - | 59.8 |
| ArmoRM-8B-v0.1 | 66.0 | 71.0 | 57.0 | 54.0 | 58.0 | 61.2 |
| InternLM2-20B-Reward | 68.0 | 70.0 | 57.0 | 58.0 | 62.0 | 63.0 |
| LLaMA-3.1-70b-Instruct | - | - | - | - | - | 59.2 |
| Claude-3.5-sonnet | 66.0 | 63.0 | 56.0 | 52.0 | 57.0 | 58.8 |
| Nemotron-4-340B-Reward | 70.0 | 65.0 | 57.0 | 49.0 | 63.0 | 60.8 |
| GPT-4o | - | - | - | - | - | 57.6 |
| *Reproduced Results of Baseline Methods From DeepSeek* | | | | | | |
| LLM-as-a-Judge | 66.0 | 68.0 | 52.8 | 50.2 | 56.8 | 58.8 |
| DeepSeek-BTRM-27B | 68.8 | 73.2 | 56.8 | 68.8 | 66.0 | 66.7 |
| CLoud-Gemma-2-27B | 68.7 | 68.8 | 53.5 | 59.0 | 62.0 | 62.4 |
| DeepSeek-PairRM-27B | 68.3 | 74.7 | 55.0 | 63.1 | 62.9 | 64.8 |
| DeepSeek-GRM-27B-RFT | 64.8 | 68.7 | 55.5 | 49.0 | 60.2 | 59.6 |
| DeepSeek-GRM-27B | 64.8 | 68.8 | 55.6 | 50.1 | 59.8 | 59.8 |
| *Results of Our Method* | | | | | | |
| **OPRM-Qwen2.5-7B** | 65.2 | 70.1 | 56.3 | 59.0 | 56.1 | 61.3 |
| **OPRM-Qwen2.5-14B** | 66.7 | 70.7 | 57.1 | 67.4 | 59.5 | 64.3 |
| **OPRM-Qwen2.5-32B** | 71.2 | 73.2 | 57.9 | 66.2 | 62.2 | 66.1 |
| **OPRM-Qwen2.5-72B** | 73.4 | 75.9 | 58.6 | 54.1 | 59.5 | 64.3 |
| *Results of Our Method (w/ Region Flooding Tuning)* | | | | | | |
| **OPRM-RgFT-Qwen2.5-7B** | 64.8 | 71.6 | 55.9 | 63.0 | 56.8 | 62.4 |
| **OPRM-RgFT-Qwen2.5-14B** | 69.5 | 74.0 | 57.3 | 67.0 | 60.0 | 65.6 |
| **OPRM-RgFT-Qwen2.5-32B** | 73.3 | 76.8 | 58.5 | 67.2 | 60.6 | 67.3 |
| **OPRM-RgFT-Qwen2.5-72B** | 72.8 | 77.1 | 59.0 | 62.0 | 61.2 | 66.4 |

Table 10: Detailed results of different methods on the RMB benchmark.

| Method | Helpfulness BoN | Helpfulness Pair | Harmlessness BoN | Harmlessness Pair | RMB |
|---|---|---|---|---|---|
| *Reported Results of Public Models* | | | | | |
| Skywork-Reward-Gemma-2-27B | 47.2 | 65.3 | 56.1 | 72.1 | 60.2 |
| DeepSeek-V2.5-0905 | - | - | - | - | 65.7 |
| Gemini-1.5-Pro | 53.6 | 76.3 | 29.9 | 66.1 | 56.5 |
| ArmoRM-8B-v0.1 | 63.6 | 78.7 | 49.7 | 66.3 | 64.6 |
| InternLM2-20B-Reward | 58.5 | 76.3 | 49.9 | 67.0 | 62.9 |
| LLaMA-3.1-70b-Instruct | 64.8 | 81.1 | 55.8 | 73.9 | 68.9 |
| Claude-3.5-sonnet | 70.5 | 83.8 | 51.8 | 76.4 | 70.6 |
| Nemotron-4-340B-Reward | - | - | - | - | 69.9 |
| GPT-4o | 63.9 | 81.5 | 68.2 | 81.4 | 73.8 |
| *Reproduced Results of Baseline Methods From DeepSeek* | | | | | |
| LLM-as-a-Judge | 55.8 | 78.5 | 50.8 | 73.9 | 64.8 |
| DeepSeek-BTRM-27B | 64.0 | 83.0 | 33.6 | 51.0 | 57.9 |
| CLoud-Gemma-2-27B | 64.7 | 81.1 | 41.7 | 66.1 | 63.4 |
| DeepSeek-PairRM-27B | 59.9 | 83.3 | 34.1 | 55.5 | 58.2 |
| DeepSeek-GRM-27B-RFT | 58.4 | 79.3 | 54.2 | 76.0 | 67.0 |
| DeepSeek-GRM-27B | 62.3 | 80.5 | 57.0 | 76.1 | 69.0 |
| *Results of Our Method* | | | | | |
| **OPRM-Qwen2.5-7B** | 63.1 | 78.4 | 65.7 | 78.8 | 71.5 |
| **OPRM-Qwen2.5-14B** | 65.8 | 80.7 | 68.2 | 80.5 | 73.8 |
| **OPRM-Qwen2.5-32B** | 69.2 | 82.1 | 68.9 | 82.0 | 75.6 |
| **OPRM-Qwen2.5-72B** | 68.7 | 82.4 | 64.2 | 78.5 | 73.5 |
| *Results of Our Method (w/ Region Flooding Tuning)* | | | | | |
| **OPRM-RgFT-Qwen2.5-7B** | 63.4 | 79.0 | 62.4 | 75.6 | 70.1 |
| **OPRM-RgFT-Qwen2.5-14B** | 66.3 | 81.3 | 65.3 | 78.2 | 72.8 |
| **OPRM-RgFT-Qwen2.5-32B** | 67.6 | 81.4 | 69.1 | 81.2 | 74.8 |
| **OPRM-RgFT-Qwen2.5-72B** | 67.9 | 82.3 | 67.1 | 79.5 | 74.2 |

## J.2 ROLE-PLAY DATA

For the domain-specific data, we employ a team of six human experts to perform accurate quality-level annotation. The data consists of Role-Play Dialogues, for which the experts assessed each prompt-response pair against four core dimensions: **Core Role-Playing Consistency**, **Interactivity & Narrative Progression**, **Fundamental Linguistic Quality**, and **Immersion**. Based on this multi-

dimensional evaluation, they directly assigned a quality level of **good**, **normal**, or **bad** to each prompt-response pair. To ensure high annotation quality and consistency, a label was only accepted if at least two experts reached a consensus. We consider this high-fidelity annotation process to be crucial for fully leveraging the capabilities of our Region Flooding Tuning method.

### J.3 Distinguishing Annotation Error from Disagreement

Our methodology emphasizes robustness to inconsistent preference data and the ability to handle annotation disagreement. To provide clarity on our data processing pipeline, it is crucial to distinguish between two fundamentally different types of label conflicts: **Annotation Errors** (Logical Inconsistency) and **Annotation Disagreements** (Subjective Ambiguity).

**Annotation Errors (Logical Inconsistency).** Our training paradigm relies on preference pairs $(y_c, y_r)$ where the chosen response $y_c$ is preferred over the rejected response $y_r$. A semantic label configuration such as **<normal, good>** (where the chosen response has a lower semantic quality than the rejected one) constitutes a direct violation of the preference premise ($y_c \succ y_r$). We classify such cases as logical inconsistencies or annotation errors rather than subjective disagreements. Empirical analysis of our dataset reveals that these contradictions are extremely rare, accounting for less than $0.1\%$ of the total samples. Consequently, our decision to filter these instances represents a standard data preprocessing aimed at eliminating verifiable noise, rather than an evasion of challenges.

**Annotation Disagreements (Subjective Ambiguity).** In contrast, the disagreement that our probabilistic framework aims to model refers to valid variations in semantic judgment where the core preference relationship remains intact. For instance, given a valid preference pair ($y_c \succ y_r$), one annotator might assign the labels **<good, normal>**, while another might assign **<good, bad>**. Both annotations respect the ordinal constraint ($y_c$ is superior to $y_r$) but differ in the perceived margin of quality. This type of variation reflects genuine subjective ambiguity in reward assessment. Unlike point-estimate models that are forced to regress to a single mean value, OPRM is specifically designed to capture this aleatoric uncertainty by learning a full probability distribution over the reward space.

In summary, the filtering step mentioned in prior sections is strictly limited to removing logical contradictions that violate the definition of a preference pair. It does not compromise our claim of robustness; rather, it ensures that the model focuses on learning meaningful distributional patterns from legitimate subjective variations.

## K Future Work

In this section, we outline several promising directions for future research that build upon the OPRM framework, such as customized Region Flooding Tuning method and customized decoding method.

### K.1 Customized Region Flooding Tuning

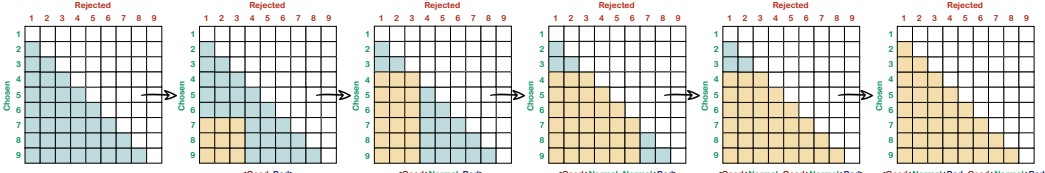

Figure 5: **Annotation Region Flooding Tuning**. As annotation ambiguity increases, the target optimization region "floods" to encompass a wider set of plausible outcomes. A more uncertain annotation results in a larger target region than a more certain one.

As discussed in Section 5.2, a core advantage of Region Flooding Tuning is its customizability. In this section, we demonstrate a novel application of this feature. Annotator inconsistency is a well-known challenge in preference data collection, an issue that is exacerbated at finer annotation granularities and leads to increased label ambiguity. To address this, we propose a method that explicitly models this ambiguity by permitting a single prompt-response pair to be associated with multiple potential

quality levels. For instance, a response might be considered both **good** and **normal**, or even all three levels in cases of extreme uncertainty. As illustrated in Figure 5, our approach handles this by optimizing over an expanded set of joint probabilities, corresponding to all plausible quality-level assignments for a given pair.

## K.2 CUSTOMIZED DECODING METHOD

The full probability distribution $p_\psi(s \mid x, y)$ produced by OPRM allows us to go beyond a simple expected score. To fully leverage this rich distributional information, we introduce **Uncertainty-aware Decoding**. This method adjusts the expected score by penalizing predictive uncertainty, thereby favoring responses that are predicted to be high-quality with high confidence. The final reward score $r_\psi(x, y)$ is calculated as:

$$r_\psi(x, y) = \underbrace{\sum_{s=a}^{b} s \cdot p_\psi(s \mid x, y)}_{\text{Expected Score}} - \underbrace{\lambda \cdot u(x, y)}_{\text{Uncertainty Term}} \tag{23}$$

where the first term is the standard expected score. The second term, $u(x, y)$, is an uncertainty measure of the distribution, such as its **Shannon entropy** or **variance**. The hyperparameter $\lambda \geq 0$ controls the strength of the uncertainty penalty.

## L COMPUTATIONAL OVERHEAD ANALYSIS

A key consideration for advanced reward modeling frameworks is the potential trade-off between performance gains and computational costs. To rigorously evaluate this, we conducted a systematic benchmark comparing the computational overhead of OPRM against a standard Bradley-Terry model under identical experimental conditions. Both models employ the Qwen2.5-32B architecture as the backbone. Benchmarks were executed using 32×NVIDIA H200 GPUs for training and 4×H200 GPUs for inference, with results reported as the average wall-clock time over three independent runs.

Table 11: Computational efficiency comparison between BTRM and OPRM.

| Method | Training Time | Inference Time | Hardware (Train) | Hardware (Infer) |
|---|---|---|---|---|
| BTRM (Standard DRM) | 7.3h | 5.5 min | $32 \times$ H200 | $4 \times$ H200 |
| **OPRM (Ours)** | **6.9h** | **2.1 min** | $32 \times$ H200 | $4 \times$ H200 |

As summarized in Table 11, OPRM exhibits training efficiency comparable to, and slightly superior to, the BTRM baseline (6.9 hours vs. 7.3 hours). This result empirically confirms that our ordinal preference modeling approach captures richer distributional information without imposing additional computational burdens during the training phase.

More notably, OPRM demonstrates a substantial advantage in inference latency, achieving a $\mathbf{2.6\times}$ speedup over BTRM (2.1 min vs. 5.5 min). This efficiency gain stems directly from OPRM's architectural design: by eliminating the need for an external scalar value head, OPRM remains strictly within the standard vocabulary space of the LLM. Consequently, it is natively compatible with highly optimized inference engines (e.g., vLLM, SGLang) without requiring custom kernel modifications or specialized implementations. This characteristic renders OPRM particularly suitable for high-throughput applications, such as online RLHF and large-scale preference ranking tasks like Best-of-N sampling, where inference speed is a critical bottleneck.

## M CASE STUDY

We present several case studies in Table 12, Table 13, and Table 14, along with the scoring results from BTRM, OPRM, and OPRM-RgFT for comparative analysis. A key finding is that by being grounded in absolute quality assessment, OPRM and OPRM-RgFT exhibit a reduced susceptibility to length bias, a known issue in purely relative models like BTRM. Moreover, for tasks with verifiable

answers, our models consistently prioritize factual correctness, whereas BTRM can be misled by other stylistic attributes.

Table 12: **Case 1**

| Example |
| --- |
| **Question**: You are a highly skilled mathematician tasked with solving complex math problems. Your goal is to provide clear, step-by-step solutions that can be easily parsed and evaluated.
Here is the math problem you need to solve:
<problem> The length of a rectangle is twice its width. Given the length of the diagonal is $5\sqrt{5}$, find the area of the rectangle. </problem>
Box your final answer using LaTeX, for example: \$x = \boxed{[Your final numerical or algebraic answer]}\$.
Now, please solve the given math problem and provide your solution in the specified format. |
| **Chosen Response**: Here's my solution to the problem:
Let the width of the rectangle be $w$. Since the length is twice the width, the length of the rectangle is $2w$.
We are given that the length of the diagonal is $5\sqrt{5}$. Using the Pythagorean theorem, we can write: $\left(5\sqrt{5}\right)^2 = w^2 + (2w)^2$
Simplifying the equation, we get: $25 \cdot 5 = w^2 + 4w^2$
Combine like terms: $25 \cdot 5 = 5w^2$
Divide both sides by 5: $25 = w^2$
Take the square root of both sides: $w = \sqrt{25} = 5$
Now that we have the width, we can find the length: $2w = 2 \cdot 5 = 10$
The area of the rectangle is the product of the width and length: $A = w \cdot 2w = 5 \cdot 10 = \boxed{50}$
Therefore, the area of the rectangle is 50. |
| **Rejected Response**: To solve this problem, we can start by using the Pythagorean theorem to relate the length and width of the rectangle to the length of the diagonal: $d^2 = l^2 + w^2$
where $d$ is the length of the diagonal, $l$ is the length of the rectangle, and $w$ is the width of the rectangle. Since the length of the rectangle is twice its width, we can write: $l = 2w$
Substituting this into the Pythagorean theorem, we get: $d^2 = (2w)^2 + w^2$
Simplifying this expression, we get: $d^2 = 5w^2$
Since we are given that the length of the diagonal is $5\sqrt{5}$, we can set up the equation: $d = 5\sqrt{5}$
Substituting this into the equation above, we get: $5\sqrt{5}^2 = 5w^2$
Simplifying this expression, we get: $25 \cdot 5 = 5w^2$
Simplifying further, we get: $125 = w^2$
Taking the square root of both sides, we get: $w = \sqrt{125} = 5\sqrt{5}$
Now that we have found the width, we can find the length by substituting this value back into the equation: $l = 2w = 2(5\sqrt{5}) = 10\sqrt{5}$
The area of the rectangle is given by: $A = lw = (10\sqrt{5})(5\sqrt{5}) = 50\sqrt{5}^2 = 50 \cdot 5 = \boxed{250}$
Final Answer: The final answer is 250. |
| **BTRM-Qwen2.5-32B**
Chosen Score: 0.03
Rejected Score: -2.70 |
| **OPRM-Qwen2.5-32B (Ours)**
Chosen Score: 7.04
Rejected Score: 3.63 |
| **OPRM-RgFT-Qwen2.5-32B (Ours)**
Chosen Score: 8.06
Rejected Score: 3.13 |

Table 13: **Case 2**

| **Example** |
| --- |
| **Question**: is $(x^y)^z = (x^z)^y$? |

**Chosen Response**: Yes, the equation $(x^y)^z = (x^z)^y$ is indeed true, and it is a direct consequence of the properties of exponents in mathematics.
To see why this is true, let's consider the equation on both sides:
Left side: $(x^y)^z$
Right side: $(x^z)^y$
First, we can simplify the left side by applying the power of a power rule, which states that $(a^m)^n = a^{(mn)}$:
Left side: $(x^y)^z = x^{(yz)}$
Now, let's simplify the right side using the same power of a power rule:
Right side: $(x^z)^y = x^{(zy)}$
Now we can see that both sides of the equation are equal:
$x^{(yz)} = x^{(zy)}$
Since the bases $(x)$ are the same and the exponents $(yz)$ and $(zy)$ are also the same, we can conclude that the equation $(x^y)^z = (x^z)^y$ is true. This property holds for any real numbers $x$, $y$, and $z$, as long as $x \neq 0$ and $y$ and $z$ are non-zero.

**Rejected Response**: A classic question about exponentiation!
Unfortunately, the answer is no, $(x^y)^z$ is not necessarily equal to $(x^z)^y$ in general.
To see why, let's break it down:
$(x^y)^z$ means "raise $x$ to the power of $y$, and then raise the result to the power of $z$". This can be written as:
$(x^y)^z = x^{(y*z)}$
On the other hand, $(x^z)^y$ means "raise $x$ to the power of $z$, and then raise the result to the power of $y$". This can be written as:
$(x^z)^y = x^{(z*y)}$
As you can see, the two expressions are not equal in general, because $y*z$ is not necessarily equal to $z*y$. For example, if $x = 2$, $y = 3$, and $z = 4$, then:
$(2^3)^4 = 2^{(3*4)} = 2^{12} = 4096$
while
$(2^4)^3 = 2^{(4*3)} = 2^{12} = 4096$
But in this case, it just so happens that $y*z = z*y$, so the two expressions are equal. However, this is not always the case.
For example, if $x = 2$, $y = 2$, and $z = 3$, then:
$(2^2)^3 = 2^{(2*3)} = 2^6 = 64$
while
$(2^3)^2 = 2^{(3*2)} = 2^6 = 64$
Again, it just so happens that the two expressions are equal in this case, but this is not a general rule.
So, to summarize: $(x^y)^z$ is not necessarily equal to $(x^z)^y$, unless $y$ and $z$ commute, i.e., $y*z = z*y$.

**BTRM-Qwen2.5-32B**
Chosen Score: -0.96
Rejected Score: -0.39

**OPRM-Qwen2.5-32B (Ours)**
Chosen Score: 7.85
Rejected Score: 3.84

**OPRM-RgFT-Qwen2.5-32B (Ours)**
Chosen Score: 8.03
Rejected Score: 2.05

Table 14: **Case 3**

| Example |
| --- |

**Question**: Anita ___ some application letters to the company but they never call her.
A.has send
B.have sent
C.have send
D.have sending
E.has sent

**Chosen Response**: The correct answer is E. "has sent".
Explanation:
In this sentence, we need to use the present perfect tense, which is formed by "has/have + past participle". The past participle of the verb "send" is "sent".
A. "has send" is incorrect because "send" is the base form of the verb, not the past participle.
B. "have sent" is incorrect because the subject "Anita" is singular, so we should use "has" instead of "have".
C. "have send" is incorrect for two reasons: "have" is used with a plural subject, and "send" is not the past participle.
D. "have sending" is incorrect because "sending" is the present participle, not the past participle.
E. "has sent" is the correct form, using "has" for the singular subject "Anita" and the past participle "sent".
Therefore, the correct sentence should be: "Anita has sent some application letters to the company, but they never call her."

**Rejected Response**: The correct option is E. "has sent."
The sentence should read: "Anita has sent some application letters to the company but they never call her."
The verb "has sent" is used because the subject of the sentence, "Anita," is singular and the action of sending the letters happened in the past. The present perfect tense is used to indicate that the action was completed in the past, but its effects are still being felt in the present. Therefore, "has sent" is the appropriate form of the verb to use in this sentence.

**BTRM-Qwen2.5-32B**
Chosen Score: 0.24
Rejected Score: -1.12

**OPRM-Qwen2.5-32B (Ours)**
Chosen Score: 8.08
Rejected Score: 6.76

**OPRM-RgFT-Qwen2.5-32B (Ours)**
Chosen Score: 8.81
Rejected Score: 8.14

