# OpenReview forum: "Learning Ordinal Probabilistic Reward from Preferences"
_ICLR.cc/2026/Conference — ICLR 2026 Poster_

### Official Review · Reviewer_hXgf · 2025-10-21

**Soundness:** 3
**Presentation:** 4
**Contribution:** 3
**Rating:** 6
**Confidence:** 4

**Summary:**

The paper addresses the limitations associated with traditional methods in Reinforcement Learning from Human Feedback (RLHF), specifically Discriminative Reward Models (DRMs) and Generative Reward Models (GRMs). DRMs are known for learning reward models based on preference signals, while GRMs identify preferences using a generative language model (LLM), potentially incorporating chain-of-thought (CoT) processing. However, DRMs often present challenges in reward interpretation and threshold determination, and GRMs are unable to perform "Best of N" comparisons or provide explicit probabilistic outputs.


In response, the authors propose a novel approach termed the Ordinal Probabilistic Reward Model (OPRM). This model outputs a complete distribution of rewards, facilitating a more nuanced understanding. The term "ordinal" reflects the method's discretization of the continuous reward space into sequences to ensure tractability. The training process involves objectives that focus on the distribution's spread, optimized using a technique called Region Flooding Tuning.


For data generation, a scoring methodology is established by curating model prompts and responses with a template that translates scores into categorical labels: Bad, Normal, and Good. This dataset is then utilized to train the model with tailored objectives that leverage Region Flooding Tuning to enhance performance consistency.


The paper employs Qwen-2.5-Instruction model variants with parameters ranging from 7B to 72B. Evaluation is conducted using Reward Bench, PPE, and RMB datasets, alongside comparisons with existing DRM and GRM techniques. The findings reveal a notable improvement over baseline models and alternative methods. Additionally, the paper discusses ablation studies assessing the influences of Region Flooding and examines the correlation with human alignment metrics and also delves into the mechanics of how the gradients operate as the learning signal.

**Strengths:**

- The paper effectively addresses a feature gap in existing models by introducing a methodology that outputs a probability distribution. As a consequence, it enables benefits such as Best of N comparison and uncertainty quantification, making it suitable for tasks requiring these features.
- The paper explores advanced techniques for constructing the objective, including methods like Region Flooding, to ensure essential learning constraints are satisfied.
- It provides a thorough comparison with existing methods using relevant benchmarks, demonstrating the proposed approach's effectiveness.
- The authors conduct a mathematical analysis illustrating how the Bradley-Terry model emerges as a special case within their distribution formulation.
- The appendix includes well-chosen examples that offer qualitative insights into the method’s output.

**Weaknesses:**

Although the method effectively addresses a specific feature gap, such as Best of N comparison and uncertainty quantification, the source of its performance enhancement compared to alternative methods remains unclear. Additional interpretative analysis would be beneficial.

**Questions:**

When generating the score for a prompt-response pair from the LLM, the token probabilities already provide a distribution. How does the learned distribution differ from this existing distribution? What supplementary insights are gained beyond the data-generating distribution? Since this data generation process resembles a GRM-based scoring system, how should we interpret the performance improvement achieved through your method when applied to data generated using GRM? Could you provide clarification on this?

---

> ### Author Response · Authors · 2025-11-20
> **Official Comment by Authors**
>
> Dear Reviewer hXgf:
>
> We thank the reviewer for the useful comments and suggestions. The concerns are addressed point by point below.
>
> > **W1: The Source of Performance Enhancement**
>
> **A1:** We thank the reviewer for recognizing our method's effectiveness. We attribute the performance enhancement to three core mechanisms: (1) Probabilistic Ranking Optimization, (2) Explicit Uncertainty Modeling, and (3) Geometric Regularization against Noise. Further details can be found in our responses to **Q1 (A2)** and **Q2/Q3 (A3)**.
>
>
> > **Q1: Learned Distribution vs. Pre-trained Token Probabilities.**
>
> **A2:** We appreciate the reviewer’s inquiry regarding the fundamental distinction between the pre-trained token distribution and the learned OPRM distribution. The transformation between the two represents the core mechanism of our reward modeling process:
>
> **(1) The Pre-trained Prior (Language Modeling Distribution)**
>
> This distribution is unaligned with the task of quality assessment. While the LLM has an ordinal understanding of numbers (i.e., '7' > '6'), it has no grounded concept of what a '7' means in terms of response quality. This initial distribution is noisy and not a reliable quality signal.
>
> **(2) Learned Reward Distribution (OPRM)**
>
> OPRM transforms this linguistic prior into a calibrated value distribution. Through our optimization objective (Eq. 6), the model learns to reallocate probability mass based on preference signals. As demonstrated in our Gradient Analysis (Appendix C), the training dynamics explicitly suppress the probability mass of lower scores and shift higher scores for chosen responses (and vice versa for rejected ones). Consequently, the learned distribution encodes uncertainty regarding quality, not just next-token uncertainty. A flat distribution in OPRM signifies epistemic uncertainty about the relative quality of the response (e.g., a hard-to-judge case), whereas a peaked distribution signals high confidence. This distributional shape is a learned feature that does not exist in the base model.
>
>
> > **Q2/Q3: Interpretation of Performance Improvement with GRM Data.**
>
> **A3:** We appreciate this insightful question. While our training data is indeed annotated by LLMs (conceptually similar to a GRM teacher), the performance gains of OPRM are not merely a result of distilling the teacher’s distribution. Instead, the improvement stems from how OPRM models and optimizes the reward signal, which extracts richer structural information than standard generative approaches.
>
> **(1) Optimization Objective (Ranking vs. Generation)**
>
> Standard GRMs typically employ a Next-Token Prediction objective to mimic the teacher's scalar output. This treats the score as a deterministic classification target (e.g., forcing the model to predict the token "8"). In contrast, **OPRM** treats the reward as a latent random variable. By optimizing the **Probabilistic Ranking Objective** (Eq. 6), we do not simply force the model to reproduce the teacher's specific score token. Instead, we optimize the full probability mass over the ordinal scale such that the expected score of the chosen response is statistically significantly higher than that of the rejected one ($P(s_c > s_r)$). This allows the model to learn robust **relative preferences** from the data, which are often more reliable than the noisy absolute scalar values provided by the teacher.
>
> **(2) Robustness via Region Flooding Tuning (RgFT)**
>
> A critical factor is that **LLM-generated scalar labels contain inherent noise and bias.** If we simply train a GRM to clone these scores, the model will overfit to the specific numeric biases of the teacher. RgFT introduces a novel regularization mechanism. By anchoring the probability mass into coarse-grained regions (*Good*/*Normal*/*Bad*) rather than overfitting to specific scalar tokens, RgFT mitigates the impact of fine-grained label noise. This allows OPRM to achieve **better calibration** than the data source itself, effectively filtering out the noise in the teacher's scalar ratings while preserving the correct ordinal ranking signal.
>
> In summary, the performance improvement is derived from shifting the learning target from **imitation** (reproducing specific tokens) to **probabilistic ordinal ranking** (learning relative distributions), which is mathematically more robust to the noise inherent in LLM-generated training data.

---

### Official Review · Reviewer_6nHQ · 2025-10-22

**Soundness:** 2
**Presentation:** 2
**Contribution:** 3
**Rating:** 2
**Confidence:** 3

**Summary:**

This paper proposes the Ordinal Probabilistic Reward Model (OPRM), a novel approach for training LLM reward models. Instead of the conventional method of training a regression head to output a single scalar reward, OPRM reframes the task as learning a probability distribution over a finite set of ordinal quality ratings (specifically, 1 to 9). The key mechanism is elegant and parameter-efficient: it directly repurposes the LLM's existing language model head, taking the normalized probabilities of the numeric tokens '1' through '9' as the reward distribution. The model is trained on preference pairs ($y_c$, $y_r$) by optimizing a loss function that maximizes the probability $P(S_c > S_r)$, where $S_c$ and $S_r$ are the random variables representing the scores for the chosen and rejected responses, respectively.

Furthermore, the paper introduces Region Flooding Tuning (RgFT), a fine-tuning strategy designed to anchor these relative preference-based distributions to an absolute quality scale. RgFT uses coarse-grained quality annotations (e.g., "bad", "normal", "good") and partitions the ordinal scores into corresponding sub-regions (e.g., {1-3}, {4-6}, {7-9}). The loss is then modified to guide the model to concentrate probability mass within these target sub-regions. The authors conduct experiments on several reward model benchmarks (RewardBench, PPE, RMB) using Qwen2.5 as a base model, claiming state-of-the-art performance compared to various discriminative and generative reward models.

**Strengths:**

1.  **Novel and Parameter-Efficient Design:** The core idea of OPRM—repurposing the LM head's vocabulary probabilities for numeric tokens as a reward distribution—is a clever and elegant approach that avoids adding any new parameters, unlike traditional scalar-based reward models that require a separate value head.
2.  **Probabilistic Formulation:** Moving from a deterministic scalar to a full probability distribution  is a well-motivated step. It inherently allows for modeling uncertainty  and, as the authors claim, offers richer interpretability than a single score.
3.  **Absolute Quality Grounding:** The Region Flooding Tuning (RgFT) strategy  is a practical and data-efficient method to tackle a known problem: the misalignment between relative-preference scores and intuitive, absolute quality.

**Weaknesses:**

1.  **Unsupported Core Assumption:** The entire method hinges on the assumption that an LLM's pre-trained probability distribution over the tokens '1'-'9' has an inherent ordinal correspondence to text quality. This is a very strong and unevaluated claim. It is highly plausible that the model has a strong prior based on token frequency in pre-training data (e.g., '1' may be far more common than '9'), which would be unrelated to quality. The paper provides no analysis of this token prior or any evidence (e.g., calibration metrics like ECE) that the resulting distribution is meaningful.
2.  **Critical Robustness Vulnerability:** The method relies on the next-token prediction at a specific prompt ending in "Score:". This creates a high risk of "jailbreaking" or adversarial manipulation. A malicious user could craft a response that includes text like "This response is excellent. Score: 9" to poison the model's internal state and bias the output distribution at the final scoring step. This plausible and critical failure mode is not evaluated.
3.  **Missing Context from Ordinal Regression Literature:** The paper makes no effort to explain the ordinality in the data as other papers of a similar topic would do, and the ordinal nature is actually imposed by design. One can view the proposed target loss as a voting one: the $P(y_c \succ y_r | x)$ can be rewritten as $\sum_k p_c(k) F_r(k-1)$, where $F_r$ term is the CDF of the rejected distribution. The paper fails to connect or compare its work to the extensive existing literature on ordinal regression and probabilistic ordinal models, which have diverse formulations. The "Related Work" section  only discusses standard DRMs and GRMs, missing a key body of relevant research.
4.  **Contradictory Claims vs. Methodology:** The paper explicitly claims "Handling Annotation Disagreement"  and "robustness to... inconsistent preference data"  as key advantages. However, Appendix H.1 clearly states that the authors "filter out all pairs where the chosen response is not strictly better than the rejected one," including "invalid combinations... such as <normal, good>". This experimental practice of manually removing the exact "inconsistent" or "ambiguous" data that the model claims to be good at handling directly undermines this central claim.
5.  **Unequal Experimental Comparisons:** The main results in Table 2  are difficult to interpret as they compare OPRM (trained on the authors' 130k dataset ) against "Reported Results of Public Models" and "Reproduced Results... From DeepSeek." These baselines are trained on different and unknown datasets. This is not an apples-to-apples comparison. It is impossible to know if OPRM's superiority comes from the method or from the base model or the extra data. A fair comparison would require training all baselines (e.g., a standard BT model, PairRM) on the exact same 130k dataset. The single ablation in Figure 4(a)  is not sufficient.
6.  **Arbitrary Partitioning in RgFT:** The 3-quality-level partition in RgFT ({1-3}, {4-6}, {7-9})  is arbitrary and presented without justification. No sensitivity analysis is performed to show why this even split is optimal, or how performance changes with different partitions (e.g., {1-2}, {3-7}, {8-9}).

**Questions:**

See the weaknesses. Overall, the idea is interesting, and I believe it has potential. But there are too many concerns to accept the claims.

---

> ### Author Response · Authors · 2025-11-20
> **Official Comment by Authors (Part 1)**
>
> Dear Reviewer 6nHQ:
>
> We thank the reviewer for the useful comments and suggestions. The concerns are addressed point by point below.
>
> > **W1: Unsupported Core Assumption regarding token probability distribution.**
>
> **A1:** We agree that relying solely on the raw, pre-trained token probabilities of an LLM carries the risk of bias due to token frequency in the pre-training corpus (e.g., '1' potentially being more frequent than '9').
>
> However, we wish to clarify that our method does not assume the pre-trained distribution is perfectly calibrated to quality. Rather, we treat the pre-trained model as a semantic initialization. Our core contribution lies in the training framework (OPRM and RgFT) which explicitly realigns these token probabilities to represent ordinal quality levels, effectively overcoming pre-training frequency biases. We support this with theoretical grounding and empirical verification.
>
> **(1) Theoretical Grounding: Leveraging Semantic Priors**
>
> The premise that LLMs can leverage their semantic understanding to assign ordinal numerical scores to text quality is not a new assumption made by us, but the foundational principle of the widely adopted LLM-as-a-Judge paradigm. Extensive prior work [1,2,3] demonstrates that LLMs (even without specific reward modeling training) possess a strong inherent capability to align numerical tokens with quality when prompted. Our work builds upon this proven capability but advances it by treating the score as a probabilistic random variable rather than a deterministic output, allowing for finer-grained training and uncertainty estimation.
>
>
> [1] Judging LLM-as-a-Judge with MT-Bench and Chatbot Arena. (NeurIPS'2023)
>
> [2] G-Eval: NLG Evaluation using Gpt-4 with Better Human Alignment. (EMNLP'2023)
>
> [3] Generative Judge for Evaluating Alignment. (ICLR'2024)
>
> **(2) Empirical Verification: Calibration Analysis (ECE)**
>
> To directly address the reviewer's concern that our distributions might be dominated by token frequency priors rather than quality signals, we conduct an Expected Calibration Error (ECE) analysis.
>
> **Setup:** We evaluate model calibration on the RewardBench dataset (5,970 pairs). Following the methodology in **Appendix H**, we establish ground-truth quality labels by aggregating GPT-4o annotations and human verification, categorizing responses into three regions: **Bad** $\{1,2,3\}$, **Normal** $\{4,5,6\}$, and **Good** $\{7,8,9\}$. We calculate ECE-10 based on the model's predicted probabilities for these regions.
>
> | Model | Accuracy (%) | ECE-10 ($\downarrow$) |
> | -------- | -------- | -------- |
> | Qwen-2.5-32B (Baseline)    | 69.56   | 26.72   |
> | OPRM-32B (Ours) |  81.04    | 10.62   |
> | OPRM-RgFT-32B (Ours) |  90.90   |  5.18  |
>
> As shown in the table, the untrained baseline indeed exhibits a high ECE (26.72), confirming the reviewer's intuition that a raw model may have uncalibrated priors. However, our OPRM training significantly reduces calibration error to 10.62, and our RgFT strategy further reduces it to **5.18**.
>
>
> > **W2: Critical Robustness Vulnerability.**
>
> **A2:** We thank the reviewer for raising this critical point regarding robustness. We acknowledge that prompt injection and adversarial manipulation are significant challenges for LLMs. However, we respectfully argue that the vulnerability described does not apply to the specific threat model and operational context of Reward Models (RMs) in the alignment pipeline. We address this concern from two perspectives:
>
> * **Internal Operational Environment**: Unlike user-facing chatbots, an RM acts as an offline, backend within the RLHF loop or during data curation. External users do not interact with the RM directly. The inputs to the RM are trajectories generated by the policy model (during RL) or datasets collected for training. The prompt template (including the final "Score:" suffix) is hard-coded by the developers. The input text (prompt x and response y) is inserted into this template.
> * **Alignment with Established Paradigms**: Our approach aligns with the LLM-as-a-Judge and Generative Reward Model paradigms. These methods widely adopt prompting strategies to elicit structured judgments. The consensus in the field is that the significant gains in interpretability and reasoning capability offered by these architectures outweigh the theoretical risk of prompt injection, particularly when deployed as offline evaluation or training signals.

---

> ### Author Response · Authors · 2025-11-20
> **Official Comment by Authors (Part 2)**
>
> >**W3: Missing Context from Ordinal Regression Literature**
>
> **A3:** We sincerely thank the reviewer for highlighting the connection between our work and the broader Ordinal Regression (OR) literature. We agree that our initial manuscript focused heavily on the specific RLHF context (DRMs vs. GRMs) and missed the opportunity to ground our method in the foundational theory of ordinal classification.
>
> **(1) Mathematical Interpretation (Voting Mechanism)**
>
> We fully agree with the reviewer’s observation regarding the "voting" formulation. Our preference probability $P(y_c \succ y_r | x)$ corresponds to the formulation: $\sum_k p_c(k) F_r(k-1)$. The $F_r$ is the Cumulative Distribution Function (CDF) of the rejected response’s score distribution. This formulation interprets the preference probability as the aggregate probability that a sample from the chosen distribution strictly exceeds a sample from the rejected distribution. We will explicitly add this derivation to Section 4.1 to bridge our Probabilistic Reward Model with standard ordinal voting mechanisms.
>
> **(2) Intrinsic Ordinality of Reward Modeling**
>
> We clarify that the ordinal nature is not merely "imposed" but is **intrinsic to the task of reward modeling**. Reward modeling fundamentally aims to measure text quality, which is a continuous, ordinal latent variable (i.e., better responses rank higher). While the preference data ($y_c \succ y_r$) provides only relative signals, it reflects this underlying ordinal utility. Therefore, our design does not artificially create order where none exists; rather, it **discretizes this latent ordinal utility** into a concrete scale (tokens ‘1’-‘9’) to make it learnable by the LLM. This aligns with standard Ordinal Regression theory, specifically the Random Utility Model paradigms, where discrete ratings represent intervals on a latent continuum.
>
> **(3) Revision Plan**
>
> In the revision, we will add the following content to the "Related Work" section:
>
> Our method aligns with the Distribution Ordering Learning paradigm [1]. Traditional approaches often rely on Continuous Space Discretization, where ordinality is treated as a regression task followed by binning [2, 3]. In contrast, OPRM learns a full probability distribution over ordinal ranks. This shares conceptual similarities with Soft Label Distribution methods like SORD [4] and Unimodal-Concentrated Loss [5], which assign soft probabilities to adjacent ranks.
>
> Furthermore, our focus on capturing reward uncertainty aligns with Probabilistic Ordinal Embeddings (POE) [6]. It proposed modeling data as multivariate Gaussian distributions to capture aleatoric uncertainty and enforced ordinal constraints via triplet loss. OPRM extends this probabilistic framework to Large Language Models (LLMs). Instead of relying on continuous latent embeddings, we utilize the LLM's discrete vocabulary to construct a probability mass function. Crucially, OPRM distinguishes itself by optimizing these ordinal distributions via a pairwise objective, thereby bridging point-wise ordinal regression with the pairwise ranking requirements of RLHF.
>
> [1] A Survey on Ordinal Regression: Applications, Advances and Prospects.
>
> [2] Deep ordinal regression network for monocular depth estimation. (CVPR'2018)
>
> [3] Deep expectation of real and apparent age from a single image without facial landmarks. (IJCV'2018)
>
> [4] Soft Labels for Ordinal Regression. (CVPR'2019)
>
> [5] Unimodal-Concentrated Loss: Fully Adaptive Label Distribution Learning for Ordinal Regression. (CVPR'2022)
>
> [6] Learning Probabilistic Ordinal Embeddings for Uncertainty-Aware Regression. (CVPR'2021)

---

> ### Author Response · Authors · 2025-11-20
> **Official Comment by Authors (Part 3)**
>
> > **W4: Contradictory Claims vs. Methodology.**
>
> **A4:** We thank the reviewer for this insightful comment. It highlights a critical distinction which we wish to clarify: the difference between annotation errors and annotation disagreements.
>
> * **Annotation Errors**: Our training data is fundamentally preference data, built on the premise that the chosen response is superior to the rejected one. A label such as <Normal, Good> directly violates this premise. This is not a nuanced disagreement but a logical inconsistency, which we classify as an annotation error. These errors are extremely rare (**<0.1% of the dataset**). Therefore, filtering them is a standard data cleaning step to remove noise, not an attempt to avoid a core challenge.
> * **Annotation Disagreements**: In contrast, annotation disagreement, which our model is designed to handle, occurs when annotators agree on the preference (chosen > rejected) but differ on the absolute quality ratings. For instance, for the same preferred pair, one expert might label it <Good, Normal> while another, focusing on different criteria, might label it <Good, Bad>. Both labels respect the core preference but reflect valid, subjective differences. It is precisely this type of ambiguity that OPRM captures by learning a full probability distribution, a capability that point-estimate models lack.
>
> We acknowledge that our manuscript did not make this distinction sufficiently clear. In the revised version, we will explicitly differentiate between annotation errors and annotation disagreements in Appendix H.1 and state that our filtering was a minor cleaning step for logical contradictions only.
>
> > **W5: Unequal Experimental Comparisons.**
>
> **A5:** The reviewer correctly notes that the models listed under "Reported Results of Public Models" and "Reproduced Results... From DeepSeek" in Table 2 are trained on different datasets. To address this, we conduct additional experiments and offer the following clarifications regarding the data setups in Table 2.
>
> **(1) Reported Results of Public Models**
>
> This category includes zero-shot models like LLM-as-a-Judge (e.g., GPT-4o), and BT-style models trained on various proprietary or public datasets. For the latter, they primarily follow the standard Bradley-Terry paradigm, meaning the primary differentiator lies in the training data used. Consequently, the controlled ablation in Figure 4(a) serves as a sufficient and direct validation of our method. By comparing OPRM against the BT objective under identical data and model settings, we isolate the algorithmic contribution, demonstrating that OPRM consistently outperforms the BT paradigm used by these public baselines.
>
> **(2) Reproduced Results of Baseline Methods From DeepSeek**
>
> We would like to highlight a critical detail regarding the comparison with DeepSeek's reproduced results. Their training data is a large collection including MATH, UltraFeedback, OffsetBias, Skywork-Reward-Preference80K, and HelpSteer2-Preference. Our training data consists only of UltraFeedback and Skywork-Reward-Preference80K, making it a strict subset of the data used by DeepSeek. The fact that our OPRM models achieve superior results while being trained on a smaller dataset strongly suggests that the performance gains are attributable to the effectiveness and data efficiency of our proposed method, rather than a data advantage.
>
> **(3) Additional Experiments**
>
> To further address the reviewer's concern, we conduct a new set of experiments as suggested. We train a standard BT model and our OPRM on the exact same 130k dataset using the publicly available Gemma-2-27B as the base model. This provides another direct, apples-to-apples comparison on a different model architecture, reinforcing our claims.
>
>
> | Method | RewardBench | PPE-P | PPE-C | RMB | Overall |
> | -------- | -------- | -------- | -------- | -------- | -------- |
> | BTRM-27B     |   81.9   | 67.4     | 66.1     | 56.5     | 68.0    |
> | OPRM-27B (Ours)    | 84.3     | 65.6    | 66.4    | 70.4    | 71.7    |
>
>
> These new results provide compelling evidence for OPRM's effectiveness.
> Our OPRM-27B model achieves an overall score of 71.7, significantly outperforming the BTRM-27B baseline's score of 68.0 (**+3.7 points**). Notably, OPRM demonstrates a massive improvement of **+13.9 points** on the RMB benchmark, indicating its enhanced ability to capture nuanced human preferences compared to the standard BT model.

---

> ### Author Response · Authors · 2025-11-20
> **Official Comment by Authors (Part 4)**
>
> > **W6: Arbitrary Partitioning in RgFT.**
>
> **A6:** We appreciate the reviewers’ queries regarding the rationale behind our binning strategies and partitions. We clarify that our configuration is not arbitrary but a principled design optimizing for computational efficiency and geometric symmetry. Furthermore, we provide additional sensitivity analyses demonstrating that the method is robust to variations in these settings.
>
> **(1) Principled Basis for Design Choices**
>
> Our selection of the $1\dots9$ scale and uniform partitioning is driven by two key constraints and objectives:
> *   **Computational Efficiency (Tokenization):** We utilize the $1\dots9$ scale to maximize granularity while ensuring each score maps to a **single token**. Mainstream tokenizers treat digits $\{0,\cdots,9\}$ as single tokens, whereas values $\ge 10$ are decomposed into multiple tokens. Restricting bins to single tokens allows OPRM to compute the full score distribution in a **single forward pass** (Eq. 7), avoiding the prohibitive cost of computing joint probabilities over multi-token sequences.
> *   **Geometric Symmetry (RgFT Objective):** We employ a uniform partition ($3\times3$ regions) as a **neutral prior**. Theoretically, this uniformity aligns with the design of RgFT, allowing the probability mass to flood into a **symmetric lower triangular geometry of consistent size**. Without domain-specific knowledge suggesting that *Good* samples require finer granularity than *Bad* ones (or vice versa), a symmetric division imposes the least inductive bias, simplifying hyperparameter selection while ensuring balanced gradient pressure across quality levels.
>
> **(2) Sensitivity Analysis: Robustness to Partitions**
>
> To empirically validate that our method works independently of these specific choices, we train and evaluate the OPRM-RgFT-32B model under a different configuration to test sensitivity. While our default setting uses a uniform $1\dots9$ scale (**Bad** $\{1,2,3\}$, **Normal** $\{4,5,6\}$, **Good** $\{7,8,9\}$), the variant uses an expanded $0\dots9$ scale with **irregular boundaries** (**Bad** $\{0,1,2,3\}$, **Normal** $\{4,5\}$, **Good** $\{6,7,8,9\}$).
>
> As shown in the table below, the performance deviation is negligible ($\Delta < 0.1\%$ on Overall score), indicating that our method **does not rely on specific boundary placements or uniform partitions to achieve high performance.**
>
> | Method                                                  | RewardBench | PPE-P | PPE-C | RMB  | Overall |
> | ------------------------------------------------------- | ----------- | ----- | ----- | ---- | ------- |
> | **Bad** $\{0,1,2,3\}$, **Normal** $\{4,5\}$, **Good** $\{6,7,8,9\}$ | 89.1        | 64.1  | 67.7  | 74.4 | 73.8    |
> | **Bad** $\{1,2,3\}$, **Normal** $\{4,5,6\}$, **Good** $\{7,8,9\}$ (Ours)  | 88.9        | 64.6  | 67.3  | 74.8 | 73.9    |

---

> > ### Comment · Reviewer_6nHQ · 2025-11-21
> >
> > I appreciate the authors' comprehensive rebuttal. Although I remain cautious about the issues of prompt-injection risk and robustness to noisy annotation data, most of my concerns have been well addressed. Accordingly, I will increase my rating to 6 and look forward to the final manuscript.

---

> > > ### Author Response · Authors · 2025-11-21
> > >
> > > We sincerely thank you for taking the time to review our paper. Your continued engagement and the valuable concerns you've highlighted are greatly appreciated. We will carefully revise the final manuscript based on the constructive feedback from you and the other reviewers to ensure a comprehensive and improved presentation.

---

### Official Review · Reviewer_axbh · 2025-10-28

**Soundness:** 3
**Presentation:** 3
**Contribution:** 3
**Rating:** 8
**Confidence:** 4

**Summary:**

The paper addresses limitations of traditional generative and discriminative reward models in RLHF. The authors propose a probabilistic reward model (PRM) training paradigm that discretizes the quality score into a finite set of ordinal ratings and trains the reward model as a classifier over these regions. They further introduce a “region flooding tuning” training strategy to improve coverage and robustness around decision boundaries. Experiments across multiple benchmarks indicate improved effectiveness over baselines.

**Strengths:**

1. Presents a novel training paradigm that requires minimal architectural changes to existing neural reward models.
2. The proposed method may improve calibration and robustness compared to traditional Bradley-Terry reward models
3. Extensive experimental evaluation across diverse datasets demonstrates practical utility.

**Weaknesses:**

1.	Although framed generally, the approach appears tied to Bradley–Terry-style preference data and is primarily evaluated in pairwise preference setups. It is unclear how well the method applies to verifiable rewards (e.g., math/code with execution-based correctness) or to process-based rewards.

2.	The choice of the number of ordinal bins, boundary placement, and mapping from discrete classes back to scalar rewards (for policy optimization) may significantly affect performance. Sensitivity analyses and principled binning strategies are not clearly presented.

3.	The training and inference overhead of probabilistic binning and flooding are not fully quantified.

**Questions:**

1.	Can you explain Eq. 8 and Figure 2 in detail?

2.	How sensitive are results to the number of bins and boundary placement?

3.	Can PRM incorporate external verifiable rewards as additional ordinal classes or as targets?

4.	What is the computational overhead of training?

5.	Does the method improve robustness to reward hacking?

---

> ### Author Response · Authors · 2025-11-20
> **Official Comment by Authors (Part 1)**
>
> Dear Reviewer axbh:
>
> We thank the reviewer for the useful comments and suggestions. The concerns are addressed point by point below.
>
> > **W1/Q3: Incorporating external verifiable rewards as additional ordinal classes or as targets.**
>
> **A1:** We thank the reviewer for this insightful question and confirm that the OPRM framework is naturally suited to incorporate external verifiable rewards as targets.
>
> Unlike traditional scalar reward models, OPRM treats the reward as a probability distribution over quality scores. This allows us to naturally integrate "hard" verifiable signals as ground-truth distributions. In our current **Region Flooding Tuning (RgFT)** (Section 5.1), we already guide the model to concentrate probability mass within specific rating sub-regions (e.g., **Bad** $\{1,2,3\}$) based on quality annotations. Within this framework, verifiable rewards function as high-confidence annotations. For a coding task, if a response passes all unit tests (External Reward = 1), we can treat this as a label for the **Good** region ($\{7,8,9\}$). If it fails compilation (External Reward = 0), it is assigned to the **Bad** region ($\{1,2,3\}$).
>
> As detailed in Appendix H.1, we effectively employed a version of this strategy during data construction. For verifiable tasks (math/code), if a response was objectively incorrect (verified against ground truth), we manually forced its label to Bad, thereby training the model to map verifiable errors to the low ordinal region.
>
> > **W2/Q2: Sensitivity analysis on the number of bins and boundary placement.**
>
> **A2:** We appreciate the reviewer’s thoughtful query regarding the sensitivity of our binning strategies and boundary configurations. To address this, we conduct additional sensitivity analyses, which demonstrate that our method is robust to variations in binning and boundary placement. Furthermore, we explain that our default configuration is a principled choice optimizing for computational efficiency and geometric symmetry.
>
> **(1) Sensitivity Analysis: Robustness to Boundary and Bin Variations**
>
> We train and evaluate the OPRM-RgFT-32B model under a different configuration to test sensitivity. While our default setting uses a uniform $1\dots9$ scale (**Bad** $\{1,2,3\}$, **Normal** $\{4,5,6\}$, **Good** $\{7,8,9\}$), the variant uses an expanded $0\dots9$ scale with **irregular boundaries** (**Bad** $\{0,1,2,3\}$, **Normal** $\{4,5\}$, **Good** $\{6,7,8,9\}$).
>
> As shown in the table below, the performance deviation is negligible ($\Delta < 0.1\%$ on Overall score), indicating that our method **does not rely on specific boundary placements or uniform partitions to achieve high performance.**
>
> | Method | RewardBench | PPE-P | PPE-C | RMB | Overall |
> | -------- | -------- | -------- | -------- | -------- | -------- |
> | **Bad** $\{0,1,2,3\}$, **Normal** $\{4,5\}$, **Good** $\{6,7,8,9\}$     |   89.1   | 64.1     | 67.7     | 74.4     | 73.8     |
> | **Bad** $\{1,2,3\}$, **Normal** $\{4,5,6\}$, **Good** $\{7,8,9\}$  (Ours) | 88.9     | 64.6    | 67.3    | 74.8    | 73.9    |
>
> **(2) Principled Basis for Default Configuration**
>
> * **Computational Efficiency (Tokenization):** We utilize the $1\dots9$ scale to maximize granularity while ensuring each score maps to a **single token**. Mainstream tokenizers treat digits $\{0,\cdots,9\}$ as single tokens, whereas values $\ge 10$ are decomposed into multiple tokens. Restricting bins to single tokens allows OPRM to compute the full score distribution in a **single forward pass** (Eq. 7), avoiding the prohibitive cost of computing joint probabilities over multi-token sequences.
> * **Geometric Simplicity**: While the method is robust to irregular boundaries, we employ a uniform partition ($3\times3$ regions) as a **neutral prior**. Specifically, this uniformity aligns with the theoretical design of RgFT, allowing the probability mass to flood into a symmetric lower triangular geometry of consistent size. Without domain-specific knowledge suggesting that *Good* samples require finer granularity than *Bad* ones, this symmetric division imposes the least inductive bias, simplifying hyperparameter selection while ensuring balanced gradient pressure across different quality levels.

---

> ### Author Response · Authors · 2025-11-20
> **Official Comment by Authors (Part 2)**
>
> > **W3/Q4: Computational overhead of training.**
>
> **A3:** We appreciate the reviewer’s inquiry regarding computational efficiency. To address this, we conduct a systematic benchmark comparing the computational overhead of OPRM against the baseline BTRM under identical settings. Both models utilize Qwen2.5-32B as the backbone and were trained on 32xH200 GPUs. We report the average results over three independent runs below:
>
> | Method | Training Time | Inference Time | Hardware (Train / Infer) |
> | -------- | -------- | -------- | -------- |
> | BTRM (DRM)     | 7.3h     | 5.5 min     | 32 * H200 / 4 * H200 |
> | OPRM (Ours)      | 6.9h     | 2.1 min    | 32 * H200 / 4 * H200 |
>
> * Training Efficiency: As shown in the table, OPRM exhibits comparable, and slightly superior, training efficiency to BTRM (**6.9h vs. 7.3h**). This confirms that our ordinal preference modeling approach captures richer distributional information without incurring any additional computational burden during training.
> * Inference Efficiency: Notably, OPRM demonstrates a substantial advantage in inference speed (**2.6x faster**). Unlike BTRM, OPRM eliminates the need for an additional value head, making it natively compatible with highly optimized inference engines (e.g., vLLM, SGLang) without modification. This efficiency is critical for high-throughput applications such as Online RLHF and large-scale preference ranking.
>
>
> > **Q1: Detail explaination of Eq. 8 and Figure 2.**
>
> **A4:** We appreciate the reviewer’s request for clarification. Equation 8 and Figure 2 collectively illustrate the evolution from our base strategy, Region Tuning (RgT), to our proposed optimal strategy, Region Flooding Tuning (RgFT).
>
> **(1) Equation 8: Formalizing Region Tuning (RgT)**
>
> Eq. 8 defines the optimization objective for RgT. Unlike the standard OPRM (Eq. 6) which sums probabilities over the entire ordinal range $\{a,\cdots, b\}$, Eq. 8 constrains the probability mass to specific **quality-level support sets**, denoted as $S_{l_{\text{chosen}}}$ and $S_{l_{\text{rejected}}}$. For instance, if a response is labeled *Good*, its support set is restricted to $S_{\text{good}} = \{7, 8, 9\}$. The equation computes the probability $P(y_c \succ y_r)$ conditional on these constraints. It sums the joint probabilities $p(s_c, s_r)$ only where $s_c \in S_{l_{\text{chosen}}}$ and $s_r \in S_{l_{\text{rejected}}}$, subject to the rank constraint $\mathbb{1}(s_c > s_r)$.
>
> **(2) The Limitation of Eq. 8 (Rectangular Optimization)**
>
> Geometrically, Eq. 8 defines a **disjoint rectangular region** in the joint score space (e.g., $s_c \in \{7,8,9\}$ and $s_r \in \{1,2,3\}$). As discussed in Section 5.2 and formally proven in **Appendix C**, optimizing within such a disjoint rectangle leads to **constant partial derivatives** for all scores within the support set.
> Specifically, within the valid rectangle, the gradient with respect to any candidate score (e.g., $s_c=7$ vs. $s_c=9$) is identical. Shifting probability mass from a lower valid score to a higher valid score yields no increase in the preference probability. Consequently, the model loses the incentive to further separate the score distributions (i.e., pushing $s_c \to 9$ and $s_r \to 1$), thereby failing to maximize the preference margin.
>
> **(3) Figure 2: Region Flooding Tuning (RgFT)**
>
> Figure 2 illustrates the solution to this limitation. To restore the margin-maximizing property, we introduce **Region Flooding**, which expands the rigid rectangular regions from Eq. 8 into a **lower triangular form**. By restoring the lower triangular shape, we ensure that the partial derivatives with respect to different scores are **distinct** (rather than constant). As derived in **Proposition C.1**, in a triangular domain, the magnitude of the gradient depends on the specific score level. This effectively recovers the optimization incentive to push the probability mass of the chosen response towards the maximum score and the rejected response towards the minimum score, significantly improving the model's discriminative power.
>
> In summary, Equation 8 establishes the foundation for utilizing absolute quality labels, while Figure 2 presents the **Flooding** mechanism that modifies the optimization landscape to preserve the desirable gradient properties of probabilistic reward modeling.

---

> ### Author Response · Authors · 2025-11-20
> **Official Comment by Authors (Part 3)**
>
> > **Q5: Does the method improve robustness to reward hacking?**
>
> **A5:** We believe our OPRM inherently offers stronger robustness against reward hacking compared to traditional scalar-based models. This robustness stems from two key capabilities:
>
> **(1) Uncertainty Quantification as a Safety Mechanism**
>
> Standard DRMs output deterministic scalar scores that often mask model ambiguity, allowing policy models to exploit out-of-distribution samples that spuriously achieve high rewards. In contrast, OPRM models the reward as a full probability distribution. When the reward model encounters "hacked" responses, the resulting prediction typically exhibits higher variance rather than a confident peak. Crucially, as formally proposed in **Appendix I.2** (Eq. 22：$r_{\psi} = \mathbb{E}[s] - \lambda \cdot u(x,y)$), our framework supports Uncertainty-aware Decoding, which incorporates a penalty term for predictive uncertainty. By dynamically penalizing responses where the model is unconfident, OPRM effectively constructs a risk-averse reward signal during RL training, actively discouraging the policy from exploiting ambiguous regions of the reward landscape.
>
> **(2) Resistance to Spurious Correlations (e.g., Length Bias)**
>
> Reward hacking frequently manifests as the exploitation of spurious correlations, such as the "length bias" observed in pairwise ranking models. Traditional BT models only learn relative rankings, making them prone to favoring longer texts without an absolute anchor. OPRM, particularly with Region Flooding Tuning, anchors rewards to absolute quality standards (*Good*/*Normal*/*Bad*). As demonstrated in Appendix J, this grounding ensures the reward signal reflects intrinsic quality rather than superficial proxies like text length.

---

### Official Review · Reviewer_Tq6c · 2025-11-02

**Soundness:** 2
**Presentation:** 1
**Contribution:** 2
**Rating:** 4
**Confidence:** 4

**Summary:**

The paper proposes a Probabilistic Reward Modeling (PRM) that treats a response’s quality as an absolute reward value rather than relative value. This absolute value is modeled as a random variable and the trained model learns its distribution. Then the author bin the continuous values into 9 discrete bins, where the ordinal quality distribution ranges from 1 to 9, and optimizes the probability that the chosen response’s score exceeds the rejected one. A training strategy, Region Flooding Tuning (RgFT), further uses good/normal/bad quality labels to improve the training. The paper reports empirical results on RewardBench, PPE, and RMB by improving over current baselines.

**Strengths:**

1. Rating the response by an integer is not novel, but the paper proposes a good way of learning the integer's distribution by contrasting the overall score distribution of chosen and rejected respones. Empirical results show the efficiency.

2. RgFT training controls the shift of probability mass more precisely by using some additional information of quality ("good/normal/bad"). It makes the training more efficient.

3. Reusing the LM head probabilities over numeric tokens is easy to implement and simple yet effective.

**Weaknesses:**

1. Calibration not measured. A primary motivation is "calibrated, interpretable distributions", yet there are no metrics (e.g., ECE) in regards of calibration reported. This is a big question, as the authors claim that they are modeling the whole distribution. Accuracy isn't enough.

2. I'm not very satisfied with the presentation of this paper. Two major concerns (but not limited to them): (1) Important model/algorithms such as RgFT are not formally defined, which is confusing. Also, gradient updates of the loss can be derived in the main text as this is an important intuition. (2) The authors spend many efforts in describing how effective their method is in the abstract and the introduction, but they do not describe their model or algorithm ideas briefly. That prevents readers from acquiring the useful information. From the perspective of readers, most submissions to NeurIPS/ICML/ICLR are SOTA, but the key difference is how and why.

3. Where do the gains come from? RgFT mixes labeled and unlabeled preference data and changes the optimized region. It would help to quantify how much improvement is due to (i) access to absolute labels vs. (ii) the flooding geometry itself vs. (iii) more total training signal.

4. Limited comparisons to other distributional RMs. The related‑work section cites distributional/uncertainty‑aware RMs (e.g., PURM, quantile approaches), but comparisons in the main tables primarily cover BT‑style and generative baselines. Including at least one representative distributional RM would better show OPRM’s advantages.

**Questions:**

1. I'm curious about what the definition of "ordinal" is in your sense. For example, there is another previous paper *Reward Modeling with Ordinal Feedback: Wisdom of the Crowd*. The term "ordinal" of yours is a bit different from theirs, and you should discuss about why your discrete absolute scores are one kind of "ordinal" label.

2. Please see Weaknesses 1. Calibration measurement should be included, or the authors should delete the claim "calibrated" in their text.

3. Please see Weaknesses 2 and 3. RgFT is the core part. The authors should include more discussions on RgFT, including formal definitions and intuitions.

4. Please see Weaknesses 4. The authors should include other distributional RMs in the comparison.

---

> ### Author Response · Authors · 2025-11-20
> **Official Comment by Authors (Part 1)**
>
> Dear Reviewer Tq6c:
>
> We thank the reviewer for the useful comments and suggestions. The concerns are addressed point by point below.
>
> > **W1/Q2: Calibration not measured. The reviewer correctly points out the lack of calibration metrics (e.g., ECE), which are essential given our motivation of modeling reward distributions.**
>
> **A1:** We agree that reporting accuracy alone is insufficient to claim "calibrated" probability estimates. To substantiate our claims, we conduct additional experiments using **Expected Calibration Error (ECE)** to rigorously quantify the reliability of our model's confidence.
>
> **Experimental Setup:** We evaluate calibration on the RewardBench dataset (5,970 prompt-response pairs).Since obtaining high-fidelity ground truth distributions for fine-grained $\{1,\cdots,9\}$ ordinal ratings is inherently noisy, we compute ECE based on the semantic quality categories defined in our method (Section 5): **Bad** $\{1,2,3\}$, **Normal** $\{4,5,6\}$, and **Good** $\{7,8,9\}$. To ensure a rigorous evaluation, we annotate all 5,970 pairs with these labels using GPT-4o, followed by human-in-the-loop verification to correct potential annotation errors. We compare our OPRM and OPRM-RgFT models against the baseline Qwen-2.5-32B-Instruct (zero-shot evaluator).
>
> **Results & Analysis:**
> As shown in the table below, our method demonstrates superior calibration properties compared to the baseline.
>
> | Model | Accuracy (%) | ECE-10 ($\downarrow$) |
> | -------- | -------- | -------- |
> | Qwen-2.5-32B (Baseline)    | 69.56   | 26.72   |
> | OPRM-32B (Ours) |  81.04    | 10.62   |
> | OPRM-RgFT-32B (Ours) |  **90.90**   |  **5.18**  |
>
> * OPRM-32B reduces ECE by **60.2%** compared to the baseline, indicating that our probabilistic objective naturally encourages better alignment between predicted probabilities and empirical correctness.
> * The OPRM-RgFT-32B model achieves the lowest ECE (**5.18**), representing an **80.6%** relative reduction over the baseline. This confirms that the Region Flooding Tuning strategy not only improves accuracy but also effectively regularizes the probability distribution, preventing overconfidence in ambiguous regions.
>
> These results corroborate our motivation that OPRM learns meaningful probabilistic interpretations of reward. We will include these metrics and the calibration analysis in the final revision.
>
> > **W2: Revising the presentation of this paper.**
>
> **A2:** We sincerely appreciate your constructive feedback regarding the presentation. We agree that the current manuscript overemphasizes empirical effectiveness at the expense of technical clarity and formalization. We will take your advice to restructure the paper to prioritize the "how" and "why" of our approach.
>
> We are currently revising the manuscript and will upload the updated version shortly. The key changes include:
>
> * We will provide a rigorous mathematical definition of the Region Flooding Tuning (RgFT) algorithm in the main text (Section 5) to ensure the training objective is unambiguous.
> * We agree that the gradient derivation provides critical intuition. We will move the theoretical analysis (Proposition C.1) and the gradient dynamics from Appendix C to Section 4.2 to better explain why OPRM effectively separates score distributions.
> * We are rewriting the Abstract and Introduction to reduce the focus on SOTA claims. Instead, we will devote more space to briefly but clearly describing the technical mechanisms of OPRM (treating reward as a random variable) and the algorithmic logic of RgFT.

---

> > ### Comment · Reviewer_Tq6c · 2025-11-26
> >
> > Thank the authors for their thorough rebuttal.
> >
> > 1. I appreciate the ECE experiments and would like to see more in the future version of your paper.
> >
> > 2. I recognize the promise the authors made. I'm sorry but I just want to know when the revised draft will be ready and submitted to ICLR.
> >
> > 3. I truly appreciate the authors detailed discussion on the source of gains.
> >
> > 4. I thank the authors for clarifying the definition of ``ordinal'' and now realize the contribution of the work in developing a model for discrete score-based rating system.
> >
> > To summarize, I truly appreciate the authors' efforts in clarifying my concerns and are definitely glad to see a more polished version in the future. The rating is now increased to 6.

---

> > > ### Author Response · Authors · 2025-11-27
> > >
> > > Thank you so much for taking the time to share your feedback! We're glad to hear that your concerns have been addressed.
> > >
> > > To answer your question regarding the timeline: we are in the process of finalizing the revision to include suggestions from all reviewers. We aim to submit the revised draft in the next few days, prior to the end of the discussion phase.
> > >
> > > If you have other comments, we are happy to address them to polish this work.

---

> ### Author Response · Authors · 2025-11-20
> **Official Comment by Authors (Part 2)**
>
> > **W3/Q3: Clarifying the Source of Gains in RgFT (Absolute Labels vs. Flooding geometry vs. Training Signal)**
>
> **A3:** We agree that quantifying the contributions of these three factors is essential for understanding the mechanism of RgFT. Below, we provide a detailed attribution of the gains based on our ablation studies and new experiments.
>
> **(i) Impact of Access to Absolute Labels (vs. RgFT)**
>
> To determine how much gain comes solely from accessing absolute quality labels (independent of our OPRM/RgFT framework), we conduct the suggested new experiments using standard classification approaches on the same dataset (combining preference data with absolute labels):
>
> * Baseline A (**Hard Classification**): A 3-way classification model (*Good*/*Normal*/*Bad*). During inference, we select the response predicted to belong to the most favorable class (via argmax).
> * Baseline B (**Scalar-Weighted Classification**): To allow for finer-grained ranking, we compute a scalar score as the expectation over class probabilities: $Score = \sum_{c \in \{G, N, B\}} P(c) \cdot \text{Value}(c)$, where values are set to the centers of our OPRM regions (e.g., 8, 5, 2).
>
> | Method | RewardBench | PPE-P | PPE-C | RMB | Overall |
> | -------- | -------- | -------- | -------- | -------- | -------- |
> | Baseline A (Hard Classification)     |   71.0   | 43.9     | 46.6     | 55.7     | 54.3     |
> | Baseline B (Scalar-Weighted Classification)     |   85.4   | 61.5     | 64.3     | 71.3     | 70.6     |
> | OPRM-RgFT-32B (Ours)    | **88.9**     | **64.6**    | **67.3**    | **74.8**    | **73.9**    |
>
> OPRM-RgFT significantly outperforms both. This confirms that the gain is not merely due to introduce the absolute labels. Instead, RgFT effectively uses these labels to anchor the preference distribution, combining the benefits of discriminative ranking (from preference data) and absolute calibration (from labels).
>
> **(ii) Impact of the Flooding Geometry**
>
> This factor isolates the benefit of the shape of the optimization region (triangular flooding) versus a naive region constraint. As detailed in **Section 6.4** and **Figure 4c** of our paper, we compare RgFT (which uses the proposed lower-triangular geometry) against a variant without flooding (which simply constrains scores to square regions). Removing the flooding geometry results in a performance drop of **1.0% to 2.9%** across benchmarks. This quantifies the gain from the geometry itself. The flooding mechanism is crucial because it preserves the gradient flow for the chosen response to achieve higher scores within its region, preventing the vanishing gradient problem inherent in rigid square constraints (as proven in **Appendix C**).
>
> **(iii) Impact of More Total Training Signal**
>
> To demonstrate that gains are not merely due to increased training signal quantity, we control for the training signal by comparing against the **BT Model - w/ Margin** (see **Table 4** and **Appendix E**). Both models utilize the full extent of the annotations (absolute labels + preferences). The **BT Model - w/ Margin** explicitly maps label combinations (e.g., <Good, Normal>) to specific scalar margins. Despite accessing identical supervision signals, OPRM-RgFT outperforms this baseline significantly (**65.4 vs. 64.2**). This result proves that the superiority of RgFT stems from its probabilistic formulation, which exploits the training signal more effectively than simply injecting it as a scalar margin.
>
> > **W4/Q4: Limited comparisons to other distributional RMs.**
>
> **A4:** We agree that a direct comparison with representative distributional reward models is essential to contextualize OPRM’s contribution. To address this, we conduct a comparative study against PURM [1] under a strictly controlled setting. Both models are trained using the Qwen-2.5-7B backbone on our dataset. We selecte the 7B scale for this comparison because the official PURM implementation currently lacks support for multi-node training, limiting its feasibility for larger-scale experiments. The results are presented below:
>
> | Method | RewardBench | PPE-P | PPE-C | RMB | Overall |
> | -------- | -------- | -------- | -------- | -------- | -------- |
> | PURM-7B     |   88.2   | 60.3     | 60.5     | 66.3     | 68.8     |
> | OPRM-7B (Ours)    | 87.8     | 61.1    | 61.3    | 71.5    | 70.4    |
>
> As shown in the table, OPRM-7B surpasses PURM-7B on PPE-P, PPE-C, and RMB, resulting in a higher overall score. Most notably, our method demonstrates a substantial performance margin on the RMB benchmark (**+5.2%**), highlighting OPRM's robustness in handling complex preference distributions. Furthermore, unlike PURM, OPRM demonstrates effective scalability to larger models (**up to 72B**) as detailed in our main paper, further validating the practical value of our approach.
>
> [1] Probabilistic Uncertain Reward Model.

---

> ### Author Response · Authors · 2025-11-20
> **Official Comment by Authors (Part 3)**
>
> > **Q1: The definition of "ordinal".**
>
> **A5:** We appreciate the reviewer’s insightful comment regarding the definition of "ordinal". It allows us to clarify the distinction between our approach and prior work like "Reward Modeling with Ordinal Feedback: Wisdom of the Crowd".
>
> The distinction lies in whether the "ordinal" characterizes the granularity of the supervision signal (Input) or the structure of the reward representation (Output):
>
>
> * **Ordinal Input Supervision (Prior Work)**: The work defines "ordinal" based on the input preference signal. They generalize the binary preference label ($y_c \succ y_r$) to a richer, ordered set of comparisons like (significantly better, better, tie, ...). This captures the degree of relative preference between two responses, while the underlying reward model typically remains a continuous scalar regressor.
> * **Ordinal Output Space (Our Work)**: In our paper, "ordinal" describes the structure of the reward model's output space. Instead of regressing a continuous scalar, OPRM models the reward as a discrete random variable supported on a finite, ordered set $S = \{a, \dots, b\}$. We term this "ordinal" because the discrete states strictly adhere to a monotonic ranking property (i.e., state $k+1$ is strictly superior to state $k$), yet they are distinct from continuous interval scales.
>
> **Why our discrete scores represent an "ordinal" label?** Our formulation aligns with the statistical definition of an **ordinal scale**, strictly distinguishing it from nominal classification or metric regression. Unlike independent nominal categories (e.g., Cat vs. Dog), our discrete states possess an inherent rank order (i.e., $9 > 8 > \dots > 1$) that is explicitly enforced by our optimization objective. By maximizing $P(s_c > s_r)$, the model operates on the inequality relationships between states rather than exact numerical distances, thereby learning the relative ordering of the scores. Crucially, while standard ordinal scales can be purely relative, our Region Flooding Tuning anchors these ordered states to semantic quality tiers (e.g., *Bad*, *Normal*, *Good*). This creates a calibrated ordinal scale that reflects absolute quality.
>
> We will incorporate this discussion into the Related Work section of our final manuscript to explicitly contrast **Ordinal Feedback** methods with our **Ordinal Reward Modeling** paradigm.

---

### Author Response · Authors · 2025-11-25
**General Response to All Reviewers**

We sincerely appreciate the reviewers’ insightful comments and constructive feedback on our manuscript. We are encouraged by the positive ratings and the recognition of our work's contribution. Specifically, we are delighted to learn that the reviewers found our **parameter-efficient design** (reusing the LM head) to be novel and easy to implement (Reviewers Tq6c and 6nHQ), and the **probabilistic formulation** to be well-motivated for modeling uncertainty and interpretability (Reviewers axbh, 6nHQ, and hXgf). The proposed **Region Flooding Tuning** strategy was highlighted as a practical and precise method for absolute quality grounding (Reviewers Tq6c, 6nHQ, and hXgf). Furthermore, reviewers commended the **extensive experimental evaluation** and the method's effectiveness (Reviewers Tq6c, axbh, and hXgf), as well as the solid **theoretical analysis** regarding the Bradley-Terry model (Reviewer hXgf). Based on the reviews, we provide a general response to the points raised by multiple reviewers and individual responses below to address each reviewer’s concerns.

(1) Regarding the questions about the experiments, we have taken the following actions:

*   **For Reviewers Tq6c and 6nHQ**, we addressed the concern regarding calibration. We evaluated the ECE on RewardBench. The results show that OPRM-RgFT significantly reduces ECE from 26.72 (baseline) to **5.18**, confirming that our probabilistic objective effectively aligns confidence with correctness.
*   **For Reviewer 6nHQ**, we added a controlled comparison using the Gemma-2-27B backbone. Under identical training data, OPRM achieved an overall score of **71.7**, outperforming the BTRM baseline (68.0) by **+3.7** points, further validating our method across different architectures. We also clarified the data comparison with DeepSeek, highlighting OPRM's data efficiency.
*   **For Reviewer Tq6c**, we conducted a comparative study against PURM at the 7B scale. OPRM surpassed PURM (Overall **70.4** vs 68.8), demonstrating superior handling of complex preference distributions.
*   **For Reviewers axbh and 6nHQ**, we performed a sensitivity analysis on binning strategies. The negligible performance deviation (**<0.1**) confirms our method's robustness.

(2) We have addressed the questions about the core idea and technical details as follows:

*   **For Reviewer Tq6c**, we provided a detailed attribution of gains within the Region Flooding Tuning strategy. Our ablation studies confirm that the performance improvement stems from the **triangular flooding geometry** and the probabilistic formulation, rather than just the inclusion of absolute quality labels or increased training signals.
*   **For Reviewer axbh**, we clarified the evolution from Region Tuning (Eq. 8) to Region Flooding Tuning (Fig. 2), explaining how the flooding mechanism restores gradient pressure for score separation. We also confirmed that external verifiable rewards can be naturally integrated as high-confidence target distributions.
*   **For Reviewer Tq6c**, we clarified the specific definition of "ordinal" in our framework. We distinguished our "Ordinal Output Space"  from "Ordinal Input Supervision" found in prior work. We emphasized that our method aligns with statistical ordinal scales where rank order is strictly enforced by the optimization objective.
*   **For Reviewer hXgf**, we elaborated on why OPRM outperforms standard Generative Reward Models. We explained that OPRM optimizes a **probabilistic ranking objective** ($P(s_c > s_r)$) rather than a token-imitation objective, making it more robust to the inherent noise in LLM-generated scalar labels.
*   **For Reviewer 6nHQ**, we clarified the connection between our method and Ordinal Regression theory, interpreting the preference probability as a voting mechanism between distributions.

(3) Missing or extended analysis:

*   **For Reviewer Tq6c**, we committed to restructuring the paper for better technical clarity. We will move the gradient dynamics analysis and the theoretical proposition (Appendix C) to the main text (Section 4.2) and provide a rigorous mathematical definition of the RgFT algorithm.
*   **For Reviewers axbh and 6nHQ**, we extended the discussion on robustness. We argued that OPRM offers better resistance to reward hacking (via uncertainty quantification) and clarified that prompt injection risks are mitigated in the offline Reward Model setting.
*   **For Reviewer axbh**, We benchmarked computational efficiency, showing that OPRM is comparable to BTRM in training but **2.6x faster** in inference.
*   **For Reviewer 6nHQ**, we explicitly distinguished between "annotation errors" (logical contradictions) and "annotation disagreements" (subjective ambiguity), clarifying that OPRM is specifically designed to model the latter.

We sincerely thank all the reviewers for their constructive suggestions. Please feel free to let us know if further details/explanations would be helpful.

Yours truly,

Authors of #16883

---

### Author Response · Authors · 2025-12-01
**[For Area Chair] Summary of Rebuttal and Discussion Outcomes**

Dear Area Chair,

We sincerely appreciate your willingness to take on this additional workload and thank you for your dedication. To assist your evaluation, we submit this brief, objective summary of our rebuttal interactions and preliminary results achieved during the discussion period.

**We guarantee strict adherence to ICLR's double-blind policy.** During the discussion period prior to the data leak:

* Through active discussion, our work gained further recognition, raising the average score from **5.0 to 6.5 (Nov 26, 20:44 UTC)**.
* All positive resolutions and score increases occurred on **Nov 26 (UTC)**. This is approximately **24 hours prior** to the public disclosure of the data leak (approx. Nov 27 15:00 UTC).
* We received positive feedback from some reviewers while still actively debating others, evidencing a natural, non-collusive review process.

These interactions reflect scientific rigor; the score increases were **solely the result of our substantial rebuttal efforts and the reviewers' careful deliberation**. While the rollback nullifies the recorded score increase, **we fully respect ICLR's policy adjustment**. We provide this concise summary to outline key pre-deadline developments for your quick review (**a detailed discussion log follows in the next official comment**).

| **Reviewer** | **Score**                                | **Summary of Review & Discussion**                           |
| :--- | :--- | :--- |
| **Tq6c**     | $\begin{array}{l} \textbf{4} \to \textbf{6} \cr \textbf{(Nov 26, 20:44 UTC)} \end{array}$ | Acknowledged our **"thorough rebuttal"** and **"detailed discussion on the source of gains."** Verified the contribution of our discrete score-based rating system after we clarified the "ordinal" definition. Following the addition of **calibration metrics (ECE)** and presentation improvements, the reviewer confirmed concerns were clarified and **raised the score from 4 to 6.** |
| **axbh**     | $\begin{array}{l} \textbf{8} \to \textbf{8}\end{array}$                           | Highly praised the work (giving an initial **8**), highlighting the **"novel training paradigm,"** **"minimal architectural changes,"** and **"extensive experimental evaluation across diverse datasets."** We addressed questions regarding sensitivity analysis and computational overhead in the appendix, reinforcing the method's robustness. |
| **6nHQ**     | $\begin{array}{l} \textbf{2} \to \textbf{6} \cr \textbf{(Nov 21, 06:28 UTC)} \end{array}$   | Initially concerned about core assumptions, but recognized the idea as **"interesting and potential."** Following our **"comprehensive rebuttal"**, the reviewer acknowledged that **"most of my concerns have been well addressed."** Consequently, the reviewer **significantly raised the score from 2 to 6.** |
| **hXgf**     | $\begin{array}{l} \textbf{6} \to \textbf{6}\end{array}$                           | Commended the work for **"effectively addressing a feature gap"** and the **"thorough comparison with existing methods."** We provided the requested additional interpretive analysis regarding performance sources, solidifying the reviewer's positive assessment of our **"advanced techniques"** and **"mathematical analysis."** |

**We summarize the improvements made to the paper during the rebuttal period** for your quick review (the revised manuscript has been submitted):

1. **Clarity & Context:**
   - Enhanced the **Introduction** to better describe the algorithm's core mechanics (Reviewer Tq6c).
   - Expanded **Related Work** to include a comprehensive summary of Ordinal Regression and Distribution Learning (Reviewer 6nHQ).
2. **New Experiments & Theoretical Analysis:**
   - Added theoretical analysis of **gradient dynamics in Section 4.2** to clarify intuition (Reviewer Tq6c).
   - Included **calibration metrics (ECE-10) in Section 6.3.1**, demonstrating the model's reliability (Reviewers Tq6c & 6nHQ).
   - Added **Sensitivity Analysis on Boundary and Bin Configurations in Appendix. E**, validating robustness (Reviewers axbh & 6nHQ).
   - Conducted an ablation study **Disentangling the Impact of Region Flooding from Label Augmentation in Appendix. G** to clarify performance sources (Reviewers Tq6c & hXgf).
3. **Robustness & Efficiency:**
   - Added analysis on **Distinguishing Annotation Error from Disagreement in Appendix. J.3** (Reviewer 6nHQ).
   - Detailed **Computational Overhead Analysis in Appendix. L** to address efficiency concerns (Reviewer axbh).

We have also posted a **General Response to All Reviewers** to summarize how we systematically resolved the common concerns. Furthermore, **detailed point-by-point responses** are provided under each reviewer's comment thread.

We fully understand and respect ICLR's policy adjustments in light of this incident. Thank you again for your consideration under this high workload. We hope this summary provides useful context for your recommendation.

Best regards,

The Authors

---

### Meta-Review · Area_Chair_4fwe · 2026-01-07

**Summary:**

1. No calibration metrics to prove the distributions were meaningful.
2. Some concerns about the "ordinal" assumption inherent in pre-trained LLM tokens
3. Lack of sensitivity analysis regarding the choice of bin boundaries.
4. Concern about potential computational overhead.
5. Concerns about the robustness against reward hacking and prompt injection attacks

**Reviewer Concerns:**

The authors' rebuttal addresses most of the concerns. They add ECE metrics, sensitivity analysis experiments, ablation studies to isolate the gains from the RgFT geometry versus just extra data.

There remains some minor concerns about the risk of prompt injection and robustness to noisy data.

**Reviewer Scores:**

Reviewer Tq6c: will raise score to 6 as the reviewer explicitly says so

Reviewer axbh: likely maintains the current score.

Reviewer 6nHQ: will raise score to 6 as the reviewer explicitly says so

Reviewer hXgf: likely maintains the current score.

---

### Decision · Program_Chairs · 2026-01-26

Accept (Poster)